# Structural basis for allosteric modulation of *M. tuberculosis* proteasome core particle

Madison Turner [1,5], Adwaith B. Uday [2,5], Algirdas Velyvis[1], Enrico Rennella[3], Natalie Zeytuni [2,4] ✉ & Siavash Vahidi [1] ✉

The *Mycobacterium tuberculosis* (*Mtb*) proteasome system selectively degrades damaged or misfolded proteins and is crucial for the pathogen's survival within the host. Targeting the 20S core particle (CP) offers a viable strategy for developing tuberculosis treatments. The activity of *Mtb* 20S CP, like that of its eukaryotic counterpart, is allosterically regulated, yet the specific conformations involved have not been captured in high-resolution structures to date. Here, we use single-particle electron cryomicroscopy and H/D exchange mass spectrometry to determine the *Mtb* 20S CP structure in an auto-inhibited state that is distinguished from the canonical resting state by the conformation of switch helices at the α/β interface. The rearrangement of these helices collapses the S1 pocket, effectively inhibiting substrate binding. Biochemical experiments show that the *Mtb* 20S CP activity can be altered through allosteric sites far from the active site. Our findings underscore the potential of targeting allostery to develop antituberculosis therapeutics.

Intracellular protein degradation is vital across all domains of life[1]. In eukaryotes, the ubiquitin proteasome system performs most non-lysosomal protein degradation and influences numerous cellular processes. Some bacteria, including the human pathogen *Mycobacterium tuberculosis* (*Mtb*), encode a proteasome system that selectively degrades damaged or misfolded proteins crucial for the pathogen's survival within host macrophages[2–7]. Consequently, the 20S core particle (CP), the central component of the proteasome system, has emerged as a viable target for tuberculosis treatment strategies[2,8–10].

*Mtb* 20S core particle features homoheptameric rings in a conserved $\alpha_7$-$\beta_7$-$\beta_7$-$\alpha_7$ configuration (Fig. 1a)[4,11,12]. These rings assemble to form three interconnected compartments: two antechambers formed by $\alpha_7$-$\beta_7$ and a central degradation chamber. Stacked β-rings enclose the degradation chamber where active sites containing the catalytic triad of βT1, βD17, and βK33 proteolyze substrates (Fig. 1b)[6,11–15]. The activity of the proteasome must be stringently regulated to prevent spurious protein degradation[5]. This is in part achieved through the switching of the conformation of the N terminus of the α-subunits. In the absence of stimuli, these "gating" residues extend across the axial

pore to cap each end of the 20S CP (Fig. 1c)[13,16], thereby blocking access to the degradation chamber and mitigating undesired substrate proteolysis. This closed conformation is converted to an open active form through the binding of regulatory particles (RPs)[7,11,16,17]. The substrate-bound RP docks onto the 20S CP via conserved C-terminal HbYX motifs (hydrophobic, tyrosine, any residue) that bind between adjacent α-subunits[11,12,18,19]. This triggers conformational changes within the N termini of the α-ring to open the axial gates, thereby allowing for substrate translocation into the degradation chamber[11].

Both eukaryotic and *Mtb* proteasome systems are modulated by allostery[12,14,20,21]. Eukaryotic 20S CP activation involves several steps[22] including pro-peptide cleavage at the active sites[23,24], RP assembly[25], substrate engagement, deubiquitylation, and significant conformational changes in RPs, all essential for full activation[26,27]. This process is further regulated by proteasome-associated proteins such as USP14/Ubp6 and UBE3C/Hul5[28,29]. Studies of archaeal and eukaryotic proteasomes also highlight allosteric communication between the active sites in the β-rings[30], between the active sites and the gating residues[31–34], between the active sites and the RPs[21,27,35–38], and between

[1]Department of Molecular and Cellular Biology, University of Guelph, Guelph, Ontario, Canada. [2]Department of Anatomy and Cell Biology, McGill University, Montréal, QC, Canada. [3]Department of Molecular Genetics, University of Toronto, Toronto, ON, Canada. [4]Centre de Recherche en Biologie Structurale (CRBS), Montréal, QC, Canada. [5]These authors contributed equally: Madison Turner, Adwaith B. Uday. ✉e-mail: natalie.zeytuni@mcgill.ca; svahidi@uoguelph.ca

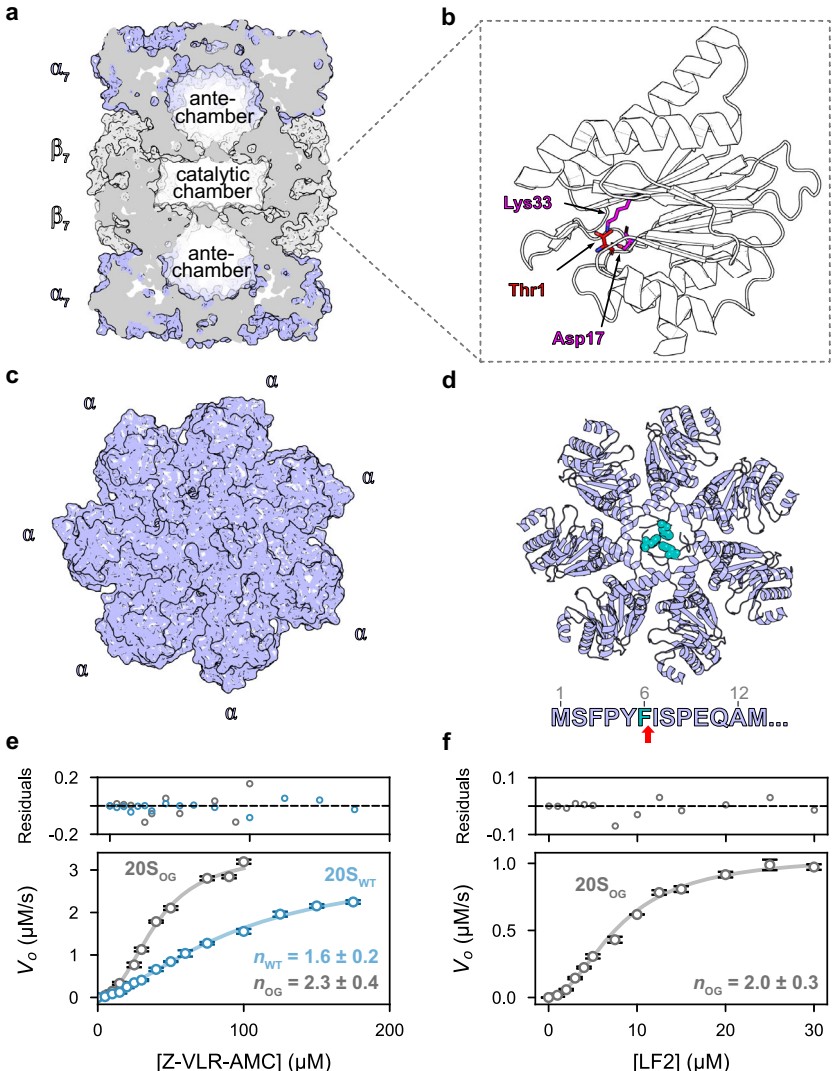

**Fig. 1 | *Mtb* 20S CP displays positive cooperativity upon substrate degradation.**
**a** The 20S CP is a barrel-shaped oligomer with α₇-β₇-β₇-α₇ architecture surrounding a central catalytic chamber flanked by two antechambers. PDB 3MI0; (**b**) Each of the β-subunits is active and contains a catalytic triad consisting of βT1 (nucleophile, red), βD17 and βK33 (magenta); (**c**) Top view of 20S CP_WT. The α-subunits function as gates that cap each end of the proteasome, effectively blocking large substrates from freely diffusing into the catalytic chamber and inhibiting spurious degradation; (**d**) F6 (teal) from three of the seven α-subunits block the central pore of the 20S CP when regulatory particles are not bound (PDB 3MKA); (**e**) Functional assays comparing the steady-state kinetics of tripeptide Z-VLR-AMC degradation between the WT and gating-defective (open-gate; OG) 20S CP variants reveal positive cooperativity during substrate hydrolysis. The *n* values from the fits to the Hill model are displayed on the plot; and (**f**) Due to the lack of gating residues, the 20S_OG variant was able to degrade the larger, 11-residue peptide LF2, with positive cooperativity. The *n* value from the fit to the Hill model is displayed on the plot. The data points and error bars in panels (**e**, **f**) represent averages and standard deviation (95% C.I.), respectively, calculated on the basis of three technical replicates (*n* = 3); Source data are provided as a Source Data file.

the α-rings spanning the entire length of the 20S CP[39,40]. The structures of *Mtb* 20S CP in isolation[4,13] and in complex with its RPs[11,19,41] have revealed many important functional features of this system. Insight into the allosteric modulation of *Mtb* proteasome was obtained in a study of the interactions between the 20S CP and its RPs, the proteasome accessory factor E (PafE; Rv3780)[14]. The 20S_βT1A variant, with a T1A mutation in the β-subunits that impairs catalytic activity (Fig. 1b), and the 20S_OG variant, featuring an open gate due to the removal of the N-terminal gating residues of the α-subunit (Fig. 1c, d), displayed a higher affinity for PafE than wildtype (WT) 20S CP[12,20]. This implies that these variants adopt a conformation that is more favourable for RP binding compared to 20S CP_WT. Yet, these conformations remain elusive in the high-resolution structures obtained to date through X-ray crystallography and electron cryomicroscopy (cryo-EM).

Here we determine near-atomic resolution cryo-EM structures of *Mtb* 20S CP, revealing an auto-inhibited state, 20S_auto-inhibited, that interconverts with the canonical resting state, 20S_resting. These structures mainly differ in the conformation of a pair of β-subunit helices at α-β interface, which we have named switch helices I and II, following the convention established by Losick and coworkers[42]. Hydrogen/deuterium exchange mass spectrometry (HDX-MS) measurements highlight a network of allosterically coupled intra- and inter-ring interactions among α- and β-subunit residues. Substitutions targeting these residues allosterically modulate the activity of the 20S CP, illuminating a pathway for the rational design of allosteric inhibitors of this critical enzyme.

## Results

### *Mtb* 20S CP displays allosteric enzyme kinetics

We individually expressed and purified α_WT, α_OG, β_WT, and β_T1A subunits, and subsequently mixed appropriate pairs of these subunits to assemble three distinct *Mtb* 20S CP variants: 20S_WT, 20S_βT1A, and 20S_OG (Supplementary Fig. 1). We verified the molecular weight of the

**Table 1 | Kinetic parameters and associated 95% confidence intervals of each 20S variant against Z-VLR-AMC**

| Variant | $K_{0.5}$ ($\mu M$) | 95% C.I. $K_{0.5}$ ($\mu M$) | $n$ | 95% C.I. $n$ | $V_{max}$ ($\mu M\,s^{-1}$) | 95% C.I. $V_{max}$ ($\mu M\,s^{-1}$) |
|---|---|---|---|---|---|---|
| $20S_{WT}$ | 94.5 | 79.9 – 111.7 | 1.6 | 1.4 – 1.7 | 3.1 | 2.8 – 3.5 |
| $20S_{OG}$ | 39.1 | 35.0 – 43.7 | 2.3 | 1.9 – 2.7 | 3.4 | 3.1 – 3.7 |
| $20S_{\alpha K52F/\beta V53Q}$ | 95.3 | 68.9 – 131.8 | 1.8 | 1.4 – 2.1 | 8.1 | 6.1 – 10.1 |
| $20S_{\beta V53Q}$ | 87.8 | 65.8 – 117.0 | 1.7 | 1.3 – 2.0 | 6.0 | 4.8 – 7.1 |
| $20S_{\alpha K52F}$ | 59.3 | 50.5 – 69.8 | 1.9 | 1.6 – 2.2 | 4.2 | 3.7 – 4.7 |
| $20S_{\alpha S17F}$ | 57.5 | 43.6 – 75.9 | 1.8 | 1.3 – 2.2 | 3.2 | 26 – 3.9 |
| $20S_{\beta Y35F}$ | 68.5 | 58.5 – 80.2 | 1.9 | 1.6 – 2.2 | 1.4 | 1.3 – 1.6 |
| $20S_{\alpha Q98K}$ | 73.8 | 59.0 – 88.5 | 2.2 | 1.5 – 2.9 | 1.0 | 0.9 – 1.2 |
| $20S_{\beta A100S}$ | – | – | – | – | – | – |
| $20S_{\beta A92-94G}$ | – | – | – | – | – | – |

Inactive variants are denoted with a dash.

individual α- and β-chains using intact protein mass spectrometry (Supplementary Fig. 2). Size exclusion chromatography coupled to multi-angle light scattering (SEC-MALS) showed that all variants had virtually identical elution profiles and estimated molecular weights consistent with the formation of the mature 20S CP (Supplementary Fig. 1e).

As in previous work[6,14,15], we examined the activity of our 20S CP variants using two substrates. Both of these contain fluorophores whose quantum yield increases upon amide bond cleavage, thereby allowing reaction monitoring via fluorescence. Z-VLR-AMC is a tripeptide that freely diffuses into the degradation chamber of the 20S CP independent of its gating function. Thus, we used this substrate to monitor the intrinsic catalytic activity of the β-subunits. Steady-state kinetic analysis of Z-VLR-AMC hydrolysis by the $20S_{WT}$ and $20S_{OG}$ variants produced sigmoidal curves, indicative of positive cooperativity between β-subunits. Fitting of the $20S_{WT}$ variant profile to the Hill model yielded a Hill coefficient of 1.6 (95% C.I. 1.4–1.7), a $K_{0.5}$ of 94.5 μM (95% C.I. 80.1–111.5), and a $V_{max}$ of 3.1 μM s⁻¹ (95% C.I. 2.8–3.5) (Fig. 1e, Table 1, and Supplementary Fig. 3). The fits for the $20S_{OG}$ profile yielded an increased Hill coefficient of 2.3 (95% C.I. 1.9–2.7), a decreased $K_{0.5}$ of 39.1 μM (95% C.I. 35.2–43.5), and unchanged $V_{max}$ of 3.4 μM s⁻¹ (95% C.I. 3.1–3.7) as compared to $20S_{WT}$ (Fig. 1e, Table 1, and Supplementary Fig. 3). The shift in the Hill coefficient and $K_{0.5}$ values between the $20S_{WT}$ and $20S_{OG}$ variants points to increased cooperative interactions when the gating residues are removed. The reaction progress curves for both the $20S_{WT}$ and $20S_{OG}$ variants display a rapid initial rate and are sensitive to changes in initial substrate concentration (Supplementary Fig. 4a, b)[43]. These observations and the microscopic length scales involved (Supplementary Fig. 4c) suggest that the changes in kinetic parameters arise from allosteric modulation of the 20S proteasome due to gate removal rather than from decreased steric hindrance caused by the absence of gating residues. Nonetheless, we concede that the role of gating residues as a diffusion barrier may also influence these changes.

Next, we used a 11-residue substrate (referred to as LF2) that is too large to freely diffuse across the α-gates. The hydrolysis rate of LF2 reports simultaneously on gating and catalytic functions of 20S CP. As anticipated, $20S_{OG}$ successfully degraded LF2, notably with a sigmoidal concentration-dependence curve (Fig. 1f). Fitting this activity profile to the Hill model yielded a Hill coefficient of 2.0 (95% C.I. 1.7–2.3), a $V_{max}$ of 1.1 μM s⁻¹ (95% C.I. 1.0–1.1) and a $K_{0.5}$ of 8.0 μM (95% C.I. 7.1–8.9) (Supplementary Fig. 5). By contrast, $20S_{WT}$ was not active against this larger substrate, confirming the expected function of the gates.

Enzyme inhibitors are often employed to stabilize specific protein conformations for detailed structural analysis. Peptidyl boronates, such as ixazomib - a clinically approved human 20S CP inhibitor - represent a class of compounds that competitively inhibit both eukaryotic and prokaryotic 20S proteasomes by mimicking the transition state that the enzyme stabilizes during substrate hydrolysis. This characteristic makes ixazomib ideal for probing the allosteric mechanisms of the *Mtb* 20S CP, as it can stabilize on-pathway conformations that are otherwise transient and elusive in structural studies[4,44,45]. We found that ixazomib inhibited the peptidase activity of $20S_{WT}$ against Z-VLR-AMC. Fitting the dose-response curve to a modified Hill model (see Supplementary Information) yielded an $IC_{50}$ value of 1.1 μM (95% C.I. 0.9–1.2) and, notably, a Hill coefficient of 2.1 (95% C.I. 1.6–2.6) (Supplementary Fig. 6a–c). We obtained similar $IC_{50}$ and Hill coefficient values for the $20S_{OG}$ variant (Supplementary Fig. 6e–g). The potency of this inhibitor enabled its use in our structural studies. Together, these results support the view that *Mtb* 20S CP is an allosteric system that interconverts between multiple conformations differing in their activity and affinity for the substrate.

## *Mtb* 20S CP adopts an auto-inhibited conformation

We used cryo-EM to experimentally determine near-atomic-resolution three-dimensional structures of four *Mtb* 20S CP variants: apo $20S_{WT}$, ixazomib-bound $20S_{WT}$, $20S_{\beta T1A}$, and $20S_{OG}$. Subsequent data processing enabled the reconstruction of high-resolution $D_7$-symmetric maps refined to global resolutions of 2.7 Å, 2.5 Å, 2.6 Å, and 2.7 Å, respectively (Supplementary Figs. 7–11, Supplementary Data 1). The resolution of all four reconstructed maps was sufficiently high to unambiguously generate molecular models of the 20S CP. The refined models displayed excellent refinement statistics and fit to the corresponding reconstructions (Supplementary Fig. 12, Supplementary Data 1).

All modelled structures adopted the canonical barrel-shaped architecture with the $\alpha_7$-$\beta_7$-$\beta_7$-$\alpha_7$ arrangement (Supplementary Figs. 8–11). Based on the enforcement of $D_7$ symmetry onto the 20S CP maps, we focused our structural analysis on a single representative α-β protomer (Fig. 2a). The density corresponding to the gating residues (αM1-I7) was not resolved in the reconstructed maps of any of the variants, likely due to their conformational flexibility. All four variants have a similar overall structure, as indicated by their low $C_\alpha$ RMSD values (0.22 – 0.63 Å) compared to apo $20S_{WT}$. Notably, $20S_{\beta T1A}$ has a slightly higher RMSD value of 0.63 Å compared to the $20S_{OG}$ variant (RMSD 0.22 Å) and the ixazomib-bound $20S_{WT}$ variant (RMSD 0.35 Å), highlighting subtle structural differences between them. A comparison of $C_\alpha$ RMSD values calculated separately for the α- and β-subunits showed that conformational changes in the β-subunit primarily account for the observed RMSD deviation in the $20S_{\beta T1A}$ variant. While the overall arrangement of the catalytic residues within the 20S CP is essentially unchanged, we noticed small but clear differences in the structure of the $20S_{\beta T1A}$ variant (Fig. 2a). Notably, the longitudinal axis of the helix formed by residues βA49-E70, hereafter referred to as switch helix I, shifts by 4 degrees in $20S_{\beta T1A}$ compared to all the other 20S variants (Fig. 2a). This results in the displacement of switch helix I by 4.7 Å, which in turn shifts an upstream β-strand and short loop containing residues βA46, βG47, βT48, and βA49, which form the back

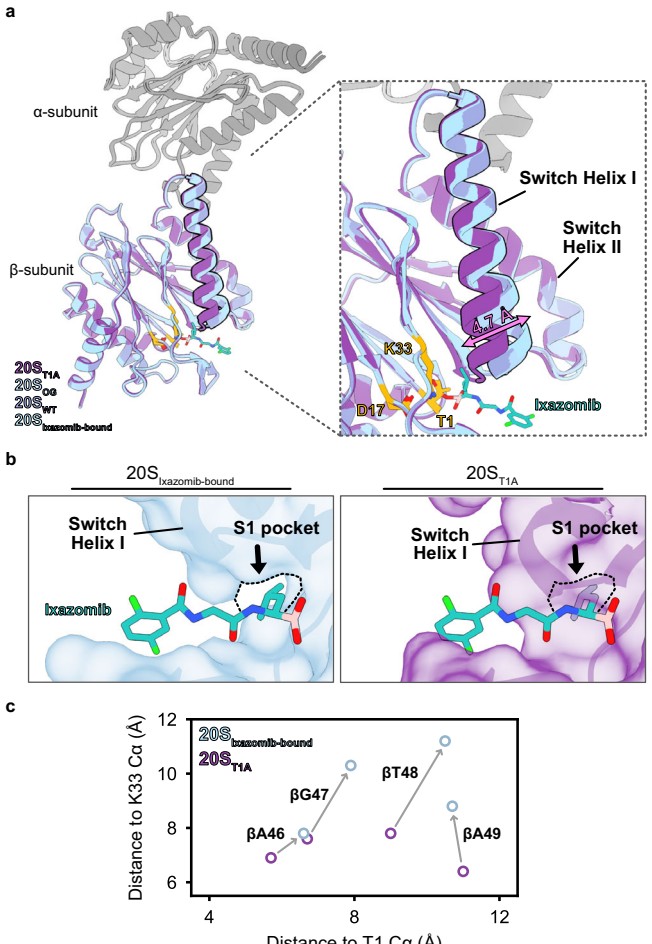

**Fig. 2 | Cryo-EM structural models reveal active and inactive conformations in the proteasome core particle. a** Superimposing one α/β-pair from each cryo-EM structure revealed little difference across the α-subunits from each variant but highlighted a key shift in the β-subunit of 20S$_{βT1A}$ where the switch helix I shifts by ~4 degrees and 4.7 Å (measured from C$_α$ of βA49) compared to the other three variants. **b** This movement shifts an upstream loop and β-strand, collapsing the S1 pocket and preventing substrate binding. **c** Residues forming the back of the S1 pocket, βA46, βG47, βT48 and βA49, move up to 3 Å between the 20S$_{WT}$ and 20S$_{βT1A}$ variants when comparing the Cα of the respective residue to the Cα of βT1 or βK33 of the catalytic triad. Source data are provided as a Source Data file.

of the S1 pocket (Fig. 2b, c). This shift in the loop leads to the S1 pocket closing completely, diminishing its volume from 260 Å³ to essentially zero, thereby effectively inhibiting substrate binding. The altered position of switch helix I is stabilized through the formation of a new hydrogen bond between the backbone carbonyl of βG47 and Nᵉ of βK33. The helix formed by residues βF76-Q96, hereafter referred to as switch helix II, is unchanged in this structure. Despite appearing unchanged here, switch helix II plays an important role in the conformational dynamics of *Mtb* 20S CP, as detailed in the subsequent sections.

For the ixazomib-bound 20S$_{WT}$ variant, we observed distinct density features that fit the inhibitor in the active sites (Supplementary Fig. 6d). Consistent with the described inhibition mechanism for this class of proteasome inhibitors[4,44,45], the boronic acid group of ixazomib is located within 1.5 Å of the catalytic oxygen atom of βT1 to allow the formation of a covalent bond. In addition, ixazomib binding is mediated by hydrogen bonding with the backbone and surrounding polar side chains (Supplementary Fig. 6d). The leucyl moiety of ixazomib sits in the deep non-polar S1 pocket, where the isobutyl side chain does not reach. Thus, the binding mode of ixazomib to *Mtb* 20S

CP is similar to that observed in its previously determined crystal structure in complex with the human 20S proteasome[3].

## A bidirectional allosteric pathway modulates *Mtb* 20S CP activity

We next used HDX-MS to probe the differences in the conformational dynamics of the four 20S CP variants studied using cryo-EM: apo 20S$_{WT}$, ixazomib-bound 20S$_{WT}$, 20S$_{βT1A}$, and 20S$_{OG}$. HDX-MS is a powerful approach for assessing protein dynamics and structure by measuring the rate at which backbone amide hydrogens exchange with the deuterium from a D$_2$O-based buffer[46,47]. The exchange rate depends on the local structure and dynamics of individual protein segments. Tightly hydrogen-bonded regions exchange slowly, while solvent-exposed and flexible regions exchange rapidly[48]. In continuous-labelling HDX-MS, a sample of protein is diluted into a D$_2$O-based buffer, and deuteration kinetics are monitored as a function of D$_2$O exposure time. Backbone amides are categorized as either "closed" (H-bonded and exchange-resistant) or "open" (solvent-accessible and exchange-competent). Closed amides can exchange only through transient conformational fluctuations, while open sites undergo HDX rapidly[49]. Ligand interactions often result in reduced deuterium uptake at residues involved in binding, while allosteric effects may manifest as increased or decreased uptake in distal regions[50]. This approach allows detailed mapping of protein-ligand interactions and conformational changes across the protein structure.

We performed bottom-up continuous labelling HDX-MS with seven D$_2$O exposure times spanning 0–1440 minutes. Peptide mapping yielded a curated list of 78 peptides covering 95% of the α-subunit sequence, and 60 peptides covering 91% of the β-subunit sequence. The average H/D back-exchange was 33% for the α- and β-subunits (Supplementary Fig. 13). We visualized our HDX-MS results as heat maps that display differential deuterium uptake across each 20S CP variant in reference to the apo 20S$_{WT}$ variant (Fig. 3). We also colour-coded the differential deuterium uptake values at $t = 10$ sec onto our 20S cryo-EM structures (Fig. 4) to help contextualize the changes in deuterium uptake. We first focused on the impact of ixazomib binding on the conformational dynamics of 20S$_{WT}$. Twelve β-subunit peptides displayed decreased deuterium uptake. These included the catalytic D17 (residues β6-25), the N terminus of switch helix I (residues β49-54), the C terminus of switch helix II (residues βF87-Q98), and the remainder belonged to regions in the immediate vicinity of the catalytic triad and the intra-ring β-β interface (Figs. 3 and 4). Of note, ixazomib binding also induced subtle changes in the conformational dynamics of the α-subunit in residues α81-91 at the α-β interface. These residues showed a decrease in deuterium uptake at longer D$_2$O exposure times.

We used the 20S$_{βT1A}$ variant to further explore the impact of perturbing the active site on the conformational dynamics of the α-subunit (Figs. 3 and 4). This single-residue substitution in the β-subunit led to differences in deuterium uptake across ten α-peptides and twelve β-peptides. In the β-subunit, peptides containing the catalytic βD17 and βK33, and those in contact with the intra-ring β-subunits (residues β6-25) showed elevated deuterium uptake relative to apo 20S$_{WT}$. Conversely, switch helices I and II (residues β43-50, β49-54, and β87-98) displayed a decrease in deuterium uptake, akin to the pattern observed with ixazomib binding. Likewise, peptides at the α-β interface showed reduced deuterium uptake, indicating protection induced by the βT1A substitution compared to the 20S$_{WT}$ variant. Notably, this protective effect in response to the βT1A substitution extends across a wider range of β-subunit peptides at the α-β interface (residues β85-91, β87-91, β87-98, β99-104, β99-110) compared to the fewer peptides affected as a result of ixazomib binding (residues β87-98). Changes in deuterium uptake extended throughout the α-subunit, including to the region immediately following the α-gates (residues α19-33) and to the structural elements involved in RP binding (Fig. 3). Inspection of

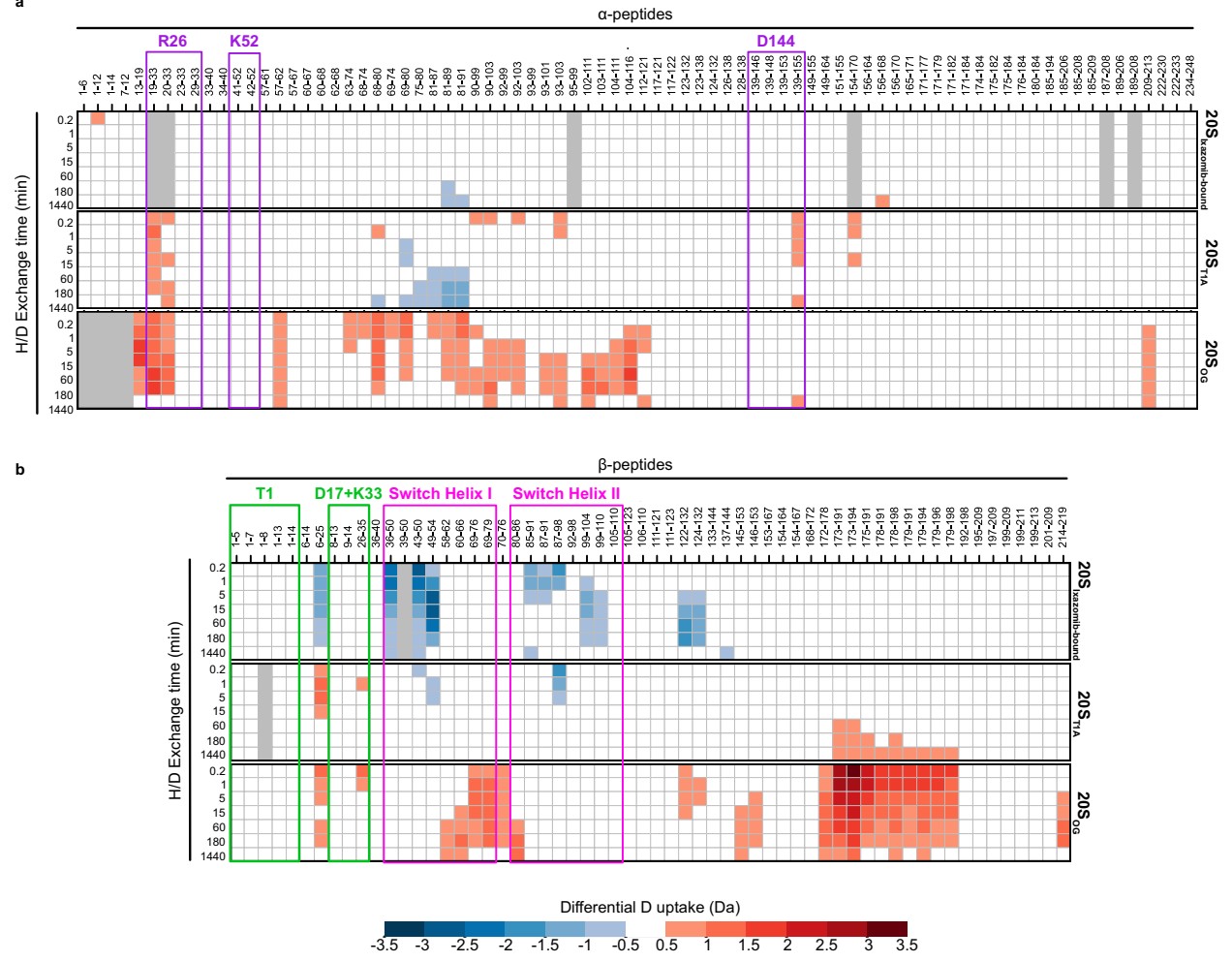

**Fig. 3 | Heat map of relative deuterium uptake highlights the allosteric pathway between the α- or β-subunits of *Mtb* 20S CP.** Changes in deuterium uptake of each state relative to the 20S$_{WT}$ variant are depicted for each peptide generated in both the (**a**) α-subunit and the (**b**) β-subunit. Only changes larger than 0.5 Da were considered significant and were coloured according to the colour bar (bottom), calculated on the basis of three technical replicates. The peptides generated for each subunit are listed across the top of the heat map and D$_2$O exposure times are listed along the side. Gray squares indicate absent data. Peptides associated with the RP-binding sites of the α-subunit and switch helices of the β-subunit are labelled.

cryo-EM structures did not show significant differences in the consensus structure in this region of the 20S CP, highlighting the unique sensitivity of HDX-MS in capturing signatures of conformational flexibility.

Finally, analysis of the differential deuterium uptake of the 20S$_{OG}$ variant relative to 20S$_{WT}$ (Figs. 3 and 4) revealed propagation of conformational changes in the opposite direction: from the α-subunit gating residues towards the β-subunit. As expected, the structural elements immediately following the deletion site in the 20S$_{OG}$ variant (residues α13-33) had increased deuterium uptake relative to 20S$_{WT}$. These differences, however, extended throughout the α-subunit and into the β-subunit. In total, twenty-two α-peptides and twenty-five β-peptides had increased deuterium uptake relative to 20S$_{WT}$. We measured elevated deuterium uptake in α-peptides proximal to the α-β interface, in those containing βD17 and βK33 of the catalytic triad, and in peptides from the switch helices I and II. While the physical removal of the gating residues might facilitate easier access for D$_2$O, the significant and sustained differences in deuterium uptake patterns and efficient diffusion over microscopic length scales are indicative of conformational changes in 20S CP rather than merely increased solvent accessibility.

Taken together, these HDX-MS results highlight regions that facilitate communication across the α-β interface into the switch helices I and II that in turn, modulate the activity of the 20S CP catalytic triad on the β-subunit ~50 Å away from the gating residues. These data also highlights the synergy and complementarity of cryo-EM and HDX-MS in probing supra-molecular systems[51]. The structure of the catalytic β-subunits of *Mtb* 20S CP in the 20S$_{OG}$ state compared to 20S CP$_{WT}$ are remarkably consistent across published work[4,11] and in our study (Fig. 2a). And yet, the HDX data unexpectedly shows significant changes in αR26 in 20S$_{βT1A}$ variant and multiple regions of the β-subunit in the 20S$_{OG}$ variant, far beyond local effects near substitution sites (Fig. 3). These data compellingly demonstrate that HDX-MS is capturing allosteric communication across the 20S CP, illustrating changes that are not merely local but indicative of broader structural interactions.

## *Mtb* 20S interconverts between a canonical resting and an auto-inhibited state on a slow timescale

Representing HDX-MS data in a heatmap format enables global mapping of conformational changes to the structure. We inspected unprocessed HDX-MS data to analyze fine details of deuterium uptake kinetics. Most peptides in the α- and β-subunits have symmetric isotopic distributions across all time points and variants, indicating they adhere to the fast timescale EX2 kinetics. However, several peptides at the α-α, α-β, and β-β interfaces display asymmetric isotopic

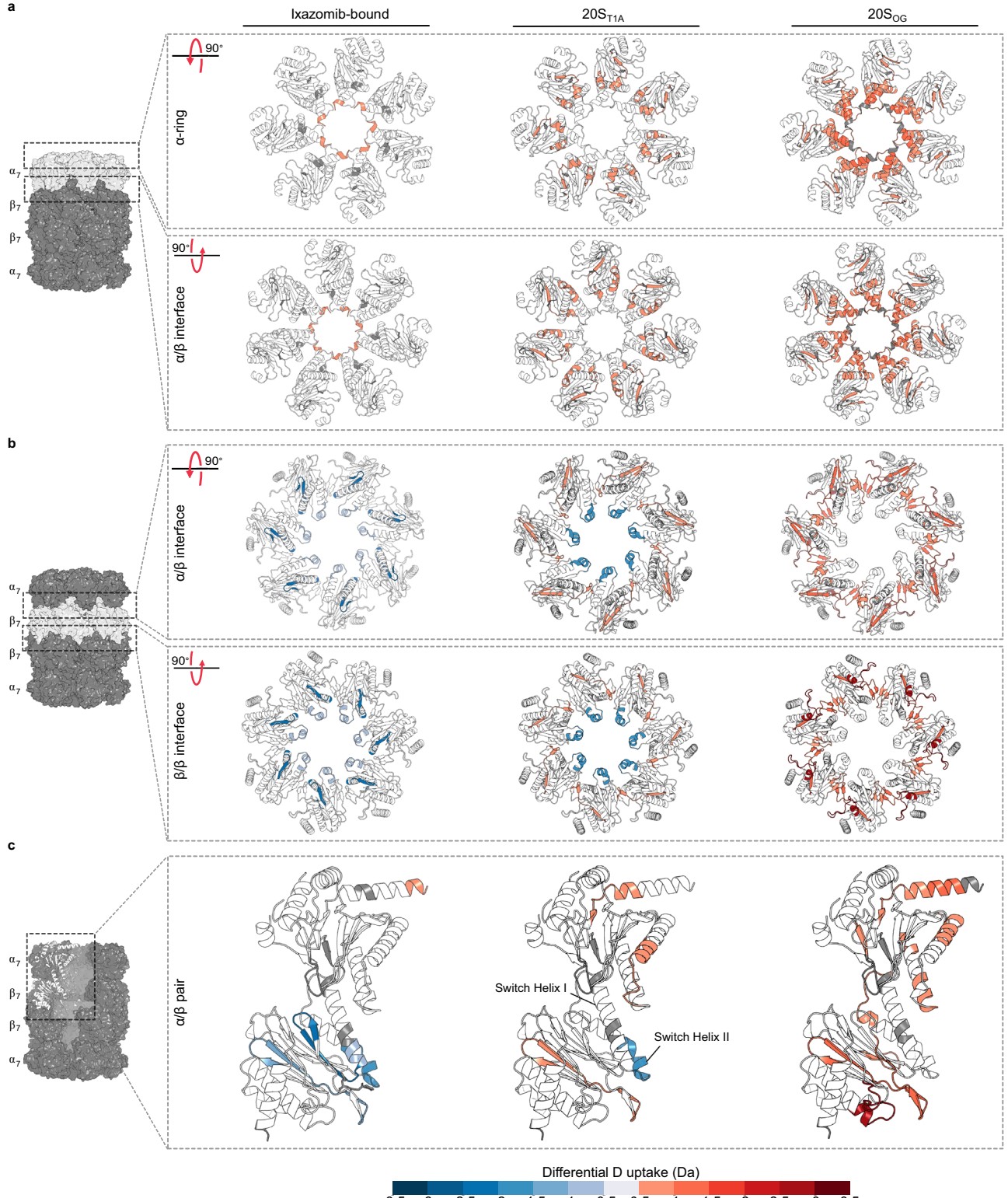

**Fig. 4 | A bidirectional pathway connects the α- and β-subunits of *Mtb* 20S CP.** Relative deuterium uptake after 10 seconds of D$_2$O exposure colour-coded onto each respective cryo-EM structure highlight regions affected in the (**a**) α-rings, (**b**) β-rings, and (**c**) across an α/β pair.

distributions typically seen in slowly interconverting systems. These include the helix immediately following the gating residues (α13-19), the structural elements that interact with the HbYX motifs at the α-α interface (α41-52, α60-67, α68-74, and α104-116), and other residues at the α-β interface (α75-80, α80-89, and α90–103), including the switch helices I and II (β69-76 and β87-98) (Figs. 3 and 4, Supplementary

Fig. 14). For example, the peptide from the C terminus of switch helix II (β87-98) clearly exhibits an asymmetric isotopic distribution in early D$_2$O exposure times in the 20S$_{βT1A}$ variant. We globally fit the isotopic distributions of this peptide with the minimum number of Gaussian curves required to achieve robust fits. Each Gaussian in these mass spectra arises from a sub-population of the protein, each with varying

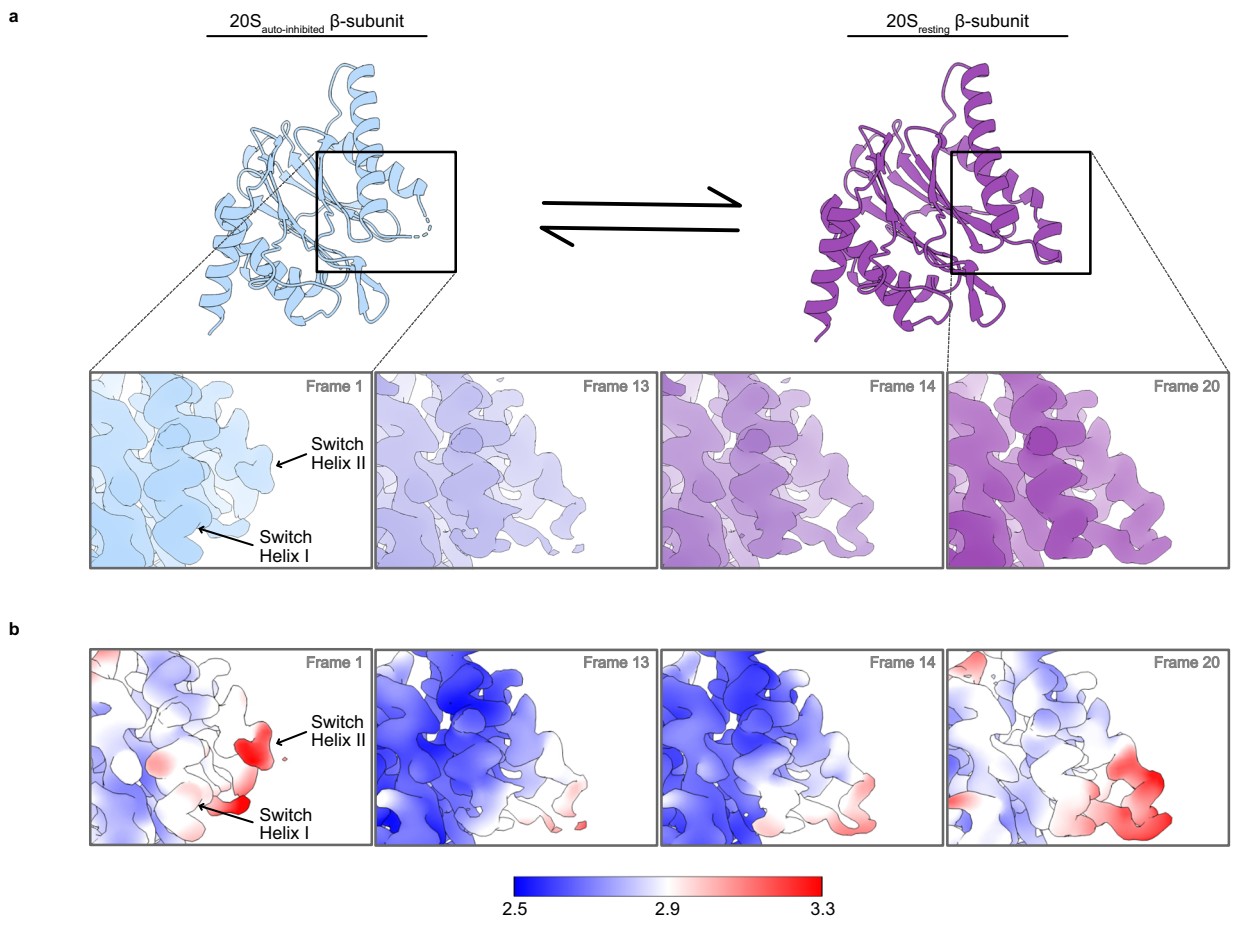

**Fig. 5 | Conformational changes in the switch helices modulate β-subunit activity. a** Cryo-EM captures the conformational changes in switch helices I and II. Switch helix I changes its position while switch helix II becomes more ordered going from frame 1 to 20, derived from 3DVA of the 20S$_{\beta T1A}$ variant. For frames 1 and 20 the refined models are presented above (20S$_{auoto-inhibited}$ and 20S$_{resting}$, respectively); and (**b**) Cryo-EM reconstructed maps of frames 1, 13, 14, and 20 of the 3DVA coloured according to the local resolution defined by the colour bar.

HDX rates at this peptide segment, thereby representing a distinct conformational state of the 20S CP. We maintained consistent width and centre for each Gaussian component across all variants and time points to minimize the number of fitting parameters, while allowing the amplitudes of the components to vary. This analysis revealed that the populations of the components in the majority of the peptides are anti-correlated while the centre of both distributions increase to higher $m/z$ values as a function of D$_2$O exposure time, consistent with mixed EX1/EX2 kinetics (Supplementary Fig. 14).

The slowly interconverting conformational states in our HDX-MS data prompted us to perform 3-Dimensional Variability Analysis (3DVA) to uncover both discrete and continuous heterogeneity across our four consensus single-particle cryo-EM maps[52]. The rapid freezing of protein samples for cryo-EM preserves structural variations, which can be uncovered using 3DVA. 3DVA deconvolutes structural heterogeneity in cryo-EM datasets into distinct modes of motion, known as principal component, each ideally representing a unique and independent conformational change, such as rotation, translation, or expansion. By identifying a principal component of interest, particles can be grouped along the principal component axis into subsets, which are then refined to produce high-resolution maps of the different conformations describing subtle and distinct structural motions[52]. 3DVA of our 20S CP datasets revealed three distinct motions within our maps. The first motion shows an alternating motion in the α-subunits, as they seemingly compete to occlude the

axial pore of the 20S CP (Supplementary Fig. 15 and Supplementary Movie 1). All variants except for 20S$_{OG}$ display this motion, likely due to the absence of the gating residues (α2-6). The second notable component of variability shows contracting and expanding of the β-rings along with rocking motions of the α-rings (Supplementary Fig. 15 and Supplementary Movie 2), a behaviour previously noted in archaeal proteasomes[52]. A similar motion is present in each 20S variant, except for the ixazomib-bound 20S$_{WT}$ variant, suggesting that inhibitor binding may reduce this type of motion in the 20S CP.

The third component of 3DVA represents the flexibility of the density corresponding to switch helices I and II and is unique to the 20S$_{\beta T1A}$ variant (Supplementary Fig. 15 and Supplementary Movie 3). Accordingly, we sorted particles from the 20Sβ$_{T1A}$ dataset into 20 intermediate reconstructions along the third principal component from 3DVA and refined them locally to produce maps with resolutions 2.5–2.7 Å (Supplementary Data 2). We subsequently fit our molecular models to the first and last intermediate reconstructed maps and achieved excellent refinement statistics (Supplementary Data 1). Frames 1 and 20 represent the most distinct states in the continuum (Supplementary Fig. 16). The comparison between the initial and final frames clearly illustrates the transition from a disordered state to a well-ordered conformation that closely mirrors with the canonical resting state of the switch helices. The initial frame closely resembles our modelled inactive consensus structure of 20S$_{\beta T1A}$ (RMSD 0.44 Å), featuring additional conformational changes in switch helix II (Fig. 5a).

The S1 pocket and the conformation of the catalytic residues are unchanged. This represents an observation of an auto-inhibited conformation of the switch helices in *Mtb* 20S CP, which we will refer to as 20S$_{auto-inhibited}$.

The final frame of 20S$_{\beta T1A}$ 3DVA, designated as 20S$_{resting}$, resembles the 20S$_{WT}$ variant structure (RMSD 0.49 Å), positioning switch helices I and II in the canonical active conformation captured previously in high-resolution structures (Fig. 5a). Inspection of sequential frames highlights the progressive conformational changes of switch helix I and the decreasing flexibility of switch helix II that accompany the resting to auto-inhibited transition. Switch helix I significantly rearranges and moves towards switch helix II. Frames 13 and 14 mark the transition state of the conformational change where both conformations of switch helix I are clearly discernible. Concurrently, residues 92-98 of switch helix II show increasing order, with their density becoming well-defined by frame 20, explaining the slower deuterium uptake of peptide β87-98 (Fig. 3). These findings are further supported by the low local resolution near the tip of switch helix II of the frames noted above and in the consensus map of the 20S$_{\beta T1A}$ variant compared to the 20S$_{WT}$ variant (Fig. 5b, Supplementary Fig. 7). Of note, switch helix I shifts by 4.7 Å in the refined atomic model of the 20S$_{\beta T1A}$ variant, while switch helix II undergoes no structural changes in the consensus map as this represents the average state of the particles in the dataset (Fig. 2a). Notably, when we applied 3DVA to both the apo 20S$_{WT}$ (Supplementary Movie 4) and 20S$_{OG}$ (Supplementary Movie 5) variants, we observed similar pattern of conformational changes as in the 20S$_{\beta T1A}$ variant, with the C terminus of switch helix II unwinding (Supplementary Fig. 17). Here switch helix I exhibits notably reduced motion compared to the 20S$_{\beta T1A}$ variant. The binding of ixazomib to 20S$_{WT}$ stabilizes an on-pathway conformation that closely mirrors 20S$_{resting}$. As such, ixazomib-bound 20S$_{WT}$ does not undergo the conformational change described above (Supplementary Movie 6). Overall, our 3DVA and HDX-MS data indicate that alternative conformation of the switch helices, though stabilized by the βT1A substitution, also naturally occurs in the wild-type 20S CP during its natural reaction cycle.

## Mutations actuate an allosteric population shift in *Mtb* 20S CP

To investigate the role of individual residues in the allosteric network of *Mtb* 20S CP, we calculated pairwise deviations between the 20S$_{auto-inhibited}$ and 20S$_{resting}$ structures and identified regions with significant conformational differences (Supplementary Fig. 18a, b). We targeted these regions using single-residue substitutions in either the α- or β-subunit. While the deviations in the α-subunit based on our atomic models are small, our HDX-MS data clearly show allosteric communication in structural elements at the α-β interface and around the RP binding site (Figs. 3 and 4). We identified a key α-subunit loop at the α-β interface that undergoes substantial rearrangement between the 20S$_{auto-inhibited}$ and 20S$_{resting}$ states. Specifically, αD93 rotates by ~80 degrees between these two states (Supplementary Fig. 18c). We noted that the αQ98K substitution would selectively stabilize the 20S$_{auto-inhibited}$ conformation of αD93, providing a potential target for modulating 20S CP activity via an α-subunit site 37 Å away from the catalytic sites. In the α-subunit, we also targeted the RP-binding pocket with two variants, 20S$_{\alpha S17F}$ and 20S$_{\alpha K52F}$, chosen based on a previous study[34] where similar substitutions in the archaeal 20S CP mimicked HbYX binding and induced gate-opening (Fig. 6). We hypothesized that these mutations facilitate communication across the α- and β-subunits, which is crucial for proteasome activation.

Structural deviations between the 20S$_{auto-inhibited}$ and 20S$_{resting}$ structures were more pronounced in the β-subunit and occurred particularly in switch helices I and II, and in adjacent structural elements (Supplementary Fig. 18b). For the β-subunit, we targeted βY35 positioned between switch helix I and the active site. In the 20S$_{\beta Y35F}$ variant, the mutation from Y to F results in the loss of a hydroxyl group

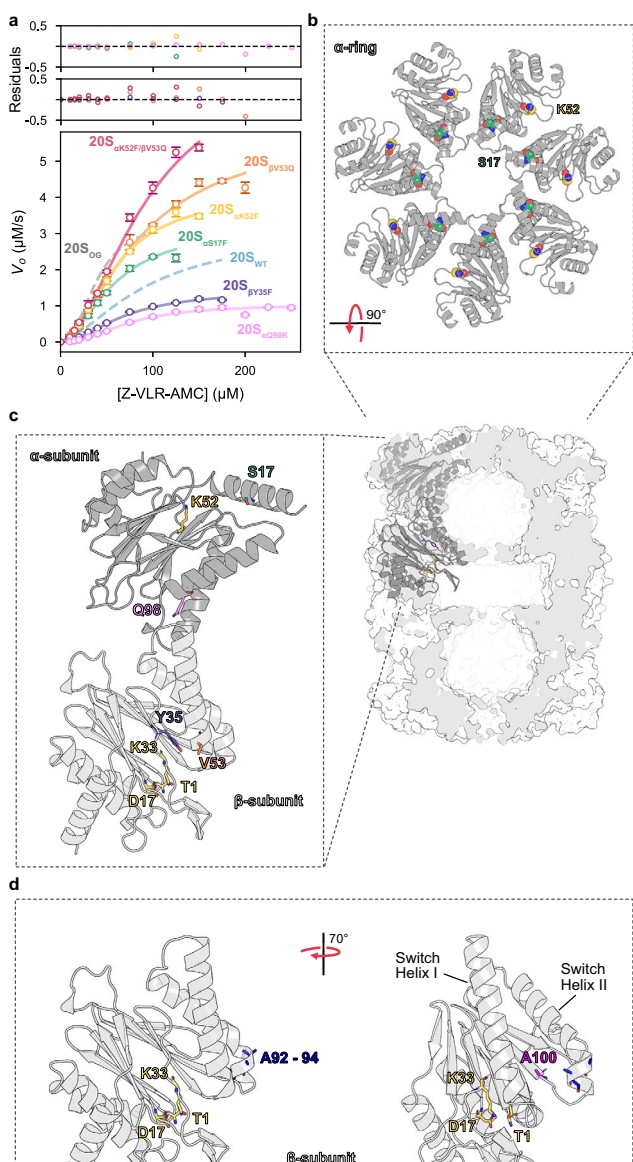

**Fig. 6 | The allosteric equilibrium of *Mtb* 20S CP can be modulated by targeting residues in either the α- or β-subunit.** Leveraging insights from both the cryo-EM and HDX-MS studies, a series of point mutations were made to target areas identified as being allosteric hotspots. **a** Single-residue substitutions to either the α- or β-subunits resulted in modulated catalytic function. The data points and error bars represent averages and standard deviation (95% C.I.), respectively, calculated on the basis of three technical replicates ($n = 3$); (**b**) Residues S17 and K53 of the α-subunits were targeted for mutation due to their role in RP-binding and proteasome activation. In a previous study using the archaeal 20S CP, the substitution of these residues with a F residue induced gate-opening; and (**c**) αQ98 was targeted as a likely allosteric modulator across the α/β-interface. βY35 and βV53 were targeted due to their proximity to the active site and switch helices. The Y35F substitution was made to prevent sidechain-mediated H-bonds that stabilize the active site. V53Q mutation was made to restore a H-bond across intra-ring β-subunits as seen in the archaeal system. **d** A92 – 94 and A100 of the β-subunits were targeted to stabilize the loop conformation of Switch Helix II. Source data are provided as a Source Data file.

essential for H-bonding with the catalytic βK33 and βV53 on switch helix I (Fig. 6). We expected that this substitution would likely weaken the interactions critical for maintaining the active conformation of the switch helices and the S1 pocket of the active site, thereby reducing the

activity of the 20S CP. Conversely, in the $20S_{\beta V53Q}$ variant would reinstate a β-β intra-ring H-bond found in the archaeal system but absent in *Mtb* 20S CP, reasoning that this would enhance activity by stabilizing the active conformation of the switch helices. Additionally, we explored the C terminus of switch helix II, which shows unwinding in $20S_{auto\text{-}inhibited}$. We hypothesized that the αQ98K substitution via long-range allosteric communication, and the βA92G-βA93G-βA94G substitution (hereafter referred to as βA92-94G), which swaps helix-stabilizing alanine residues for glycines, would both destabilize switch helix II and favour the $20S_{auto\text{-}inhibited}$ conformation and reduce activity. Lastly, βA100 is positioned at the N terminus of a β-strand between switch helices I and II in the $20S_{resting}$ structure, and has elevated RMSD between the two states (Supplementary Fig. 18b). The preceding residues are disordered in the $20S_{auto\text{-}inhibited}$ state. We sought to stabilize the disordered conformation of switch helix II with a βA100S substitution (Fig. 6). This substitution provides an additional hydrogen bond with the backbone of βL101 in the $20S_{auto\text{-}inhibited}$ conformation and was also expected to reduce activity.

We used size exclusion chromatography to purify samples of correctly assembled oligomeric particles of all 20S variants. Steady-state kinetic analysis of Z-VLR-AMC hydrolysis by the variants yielded sigmoidal curves, indicative of positive cooperativity. Fitting of the data to the Hill model yielded Hill coefficients ranging from 1.7 to 2.2 (Fig. 6, Table 1, and Supplementary Fig. 3). The $20S_{\alpha Q98K}$ variant that we designed to favour the auto-inhibited conformation, had a 3-fold reduced $V_{max}$ value of $1.0\ \mu M\ s^{-1}$ (95% C.I. 0.9–1.2) relative to $20S_{WT}$, with statistically insignificant changes in $K_{0.5}$ values (95% C.I. 59.0–88.5). The α-subunit variants, $20S_{\alpha S17F}$ and $20S_{\alpha K52F}$, showed $K_{0.5}$ values of $57.5\ \mu M$ (95% C.I. 43.6–75.9) and $59.3\ \mu M$ (95% C.I. 50.5–69.8), respectively, while the β-subunit variants, $20S_{\beta Y35F}$ and $20S_{\beta V53F}$, displayed increased $K_{0.5}$ values of $68.5\ \mu M$ (95% C.I. 58.5–80.2) and $86.9\ \mu M$ (95% C.I. 65.8–117.0), respectively. Interestingly, while the $V_{max}$ of $20S_{\alpha S17F}$ matched those of the $20S_{WT}$ and $20S_{OG}$ variants at $3.2\ \mu M\ s^{-1}$ (95% C.I. 26–3.9), the $20S_{\alpha K52F}$ and $20S_{\beta V53Q}$ variants exhibited increased $V_{max}$ values of $4.2\ \mu M\ s^{-1}$ (95% C.I. 3.7–4.7) and $5.9\ \mu M\ s^{-1}$ (95% C.I. 4.9–7.2), respectively. In contrast, the $20S_{\beta Y35F}$ variant had a reduced $V_{max}$ of $1.4\ \mu M\ s^{-1}$ (95% C.I. 1.3–1.6). The switch helix II variants $20S_{\beta A92\text{-}94G}$ and $20S_{\beta A100S}$ that we designed to favour the auto-inhibited conformation were inactive. We next combined the most active variants of the α- and β-subunits to determine if the effects of the activating mutations were additive. The $20S_{\alpha K52F/\beta V53Q}$ variant maintained unchanged $n$ and $K_{0.5}$ values compared to the other variants, yet it achieved the highest $V_{max}$, reaching $8.0\ \mu M\ s^{-1}$ (95% C.I. 6.2–10.4).

We used pulsed HDX-MS to compare the structure of several 20S variants to $20S_{WT}$ in the apo state. The deuteration pattern of the $20S_{\alpha Q98K}$ variant in the apo state was indistinguishable from that of $20S_{WT}$, except for the C terminus of switch helix II which was more deuterated compared to $20S_{WT}$ (Supplementary Fig. 19a). This is consistent with the αQ98K substitution affecting the conformation of switch helix II via long-range allosteric communication. The βA92-94G substitution also destabilized switch helix I via the loss of local interactions. The βA100S substitution had increased deuterium uptake in several regions in the α- and β-subunits, including in switch helices I and II, in peptides encompassing the catalytic residues, and in residues in the α-subunits involved in RP binding and the α-β interface. These data affirm that the substitutions exert their intended effects on the structure and dynamics of *Mtb* 20S CP. Next, we tested the ability of these variants in binding ixazomib. Similar to $20S_{WT}$, all variants showed protection in catalytic residues and in switch helix I upon ixazomib binding. The $20S_{\alpha Q98K}$ variant, which was active in our peptidase assays, also showed protection switch helix II in response to ixazomib binding, indicating that the impact of the substitution can be reversed via the addition of small molecules that favour the canonical resting state of 20S CP. However, the $20S_{\beta A92\text{-}94G}$ variant, which was

inactive in our assays, did not undergo protection upon ixazomib binding in the C-terminal portion of switch helix II, further highlighting the role of the switch helices in modulating the activity of *Mtb* 20S CP.

One of the primary roles of the proteasome system in mycobacteria is to degrade pupylated substrates[10,53,54]. Mycobacterial proteasome activator (Mpa) recruits, progressively unfolds, and threads pupylated substrates into the 20S CP for degradation. We sought to assess the ability of our most active 20S variants in degrading a Pup-dihydrofolate reductase (DHFR) fusion substrate as part of the Mpa:20S CP degradation complex. *Mtb* $20S_{WT}$ with wild-type full-length Mpa failed to degrade Pup-DHFR (Supplementary Fig. 20); only the $20S_{OG}$ variant degraded Pup-DHFR. These observations are consistent with the literature, which describes the β-grasp domain of Mpa preventing effective binding with 20S $CP_{WT}$[55]. Lastly, we tested the ability of the $20S_{\beta V53Q}$ variant in degrading Pup-DHFR. Despite its higher peptidase activity, we opted to exclude the αK52F substitution to avoid potential interference with Mpa binding. Even our second-most active variant ($20S_{\beta V53Q}$) showed no detectable activity against Pup-DHFR (Supplementary Fig. 20). While these activating substitutions could be introduced in the background of $20S_{OG}$, the effects attributable to these substitutions may be obscured due to the dominant functional characteristics of the $20S_{OG}$ variant. This complexity would hinder the clear attribution of any observed changes in activity effects directly to the substitutions, as they could be potentially influenced by the affinity of the 20S variant-Mpa interaction.

## Discussion

Elegant crystallography and cryo-EM studies of *Mtb* 20S CP in isolation[4,13] and bound to its RPs[11,19,41] have been instrumental in advancing our understanding of bacterial proteasome systems. Indeed, three of the four variants we studied by cryo-EM closely resemble the existing crystal structures of the 20S CP in its canonical resting state. The fourth variant, however, provided an example of *Mtb* 20S CP with the switch helices at the α-β interface in a conformation that completely obstructs the S1 substrate binding pocket, thereby modulating 20S activity. While the βT1A substitution stabilizes the $20S_{auto\text{-}inhibited}$ structure, we also observe conformational changes in switch helix II in the 3DVA of the wild-type enzyme. The structural plasticity of the switch helices is further substantiated by their bimodal isotopic distribution in $20S_{WT}$ HDX-MS data, and by amino acid substitutions that selectively stabilize the inactive conformation. These observations suggest that the $20S_{auto\text{-}inhibited}$ structure naturally occurs within the wild-type enzyme. In contrast to the βT1A variant structure reported by Li et al., which was crystallized with the β-propeptide present and adopts the canonical state of the 20S, our cryo-EM structure of the $20S_{T1A}$ variant lacks the β-propeptide, potentially accounting for the observed conformational differences. The crystalline environment can influence the backbone conformation, side-chain flexibility, and the arrangement of water molecules and therefore may not fully represent the diversity of conformations the 20S can adopt in solution[56].

Our attempts to explore the impact of amino acid substitutions on Mpa-mediated degradation of pupylated substrates were complicated by the weak interaction between Mpa and the full-length 20S CP. As in previous work[19,41], stable interaction and effective degradation of pupylated substrates by *Mtb* Mpa-20S were only achieved in vitro with the $20S_{OG}$ variant lacking the first seven residues of the α-subunits. The introduction of mutations into the $20S_{OG}$ variant would mask the intrinsic effects of these mutations due to its dominant functional characteristics. This could complicate the attribution of observed effects solely to the substitutions, as function might also be influenced by the affinity of the variant-Mpa interactions. To avoid these complexities, our experiments utilize a short peptidic substrate that directly probes the proteasome's active site independent of RP interactions. While this approach limits our ability to fully capture the

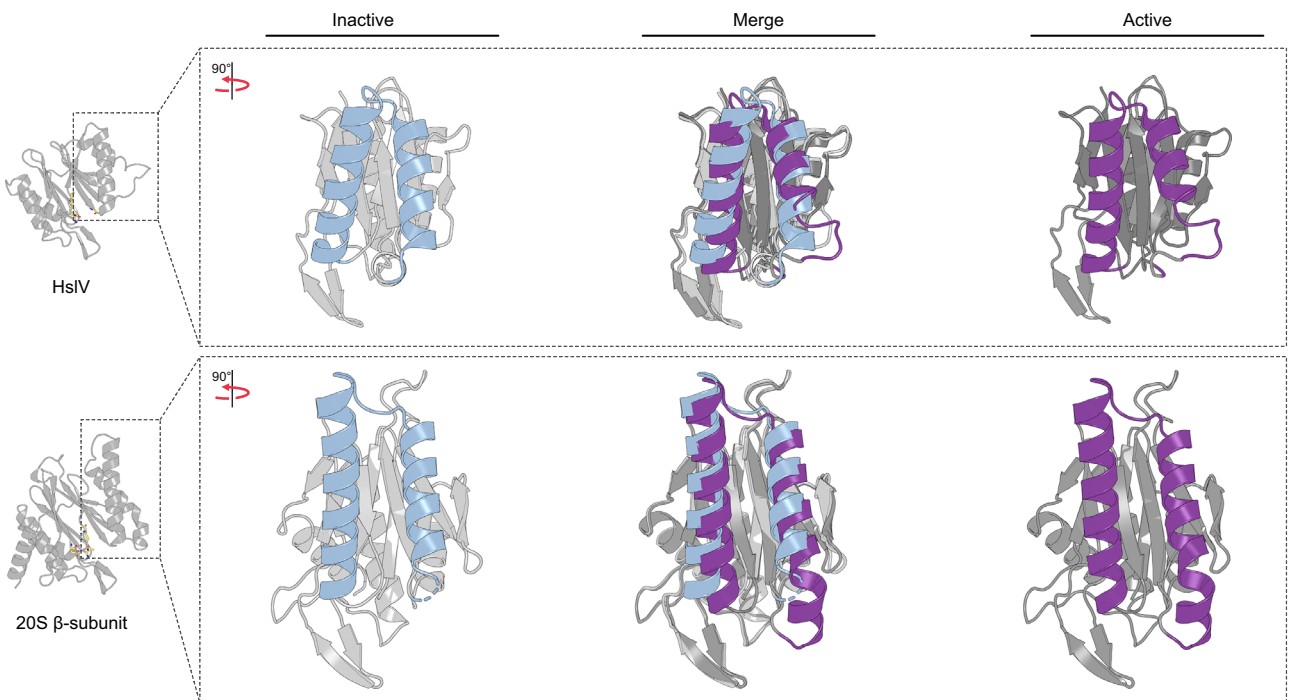

**Fig. 7 | Switch helices are conserved and regulate activity in the ancestral enzyme, HslV.** Comparison of the switch helices from the 20S β-subunit to those from HslV, an ATP-dependent protease homologous to the proteasome β-subunit, highlight conserved structural transitions that regulate proteolytic function. The C-terminal region of switch helix II undergoes an order to disordered conformational change between inactive (PDB 1G3K) and active (PDB 1G3I) states of HslV in a manner similar to that seen in the *Mtb* 20S β-subunit.

biological role of the *Mtb* proteasome, it offers more definitive evidence of allosteric modulation within the 20S CP.

Our structural insights advance our understanding of the functional dynamics of *Mtb* 20S CP. Molecules that selectively bind to and stabilize the $20S_{auto-inhibited}$ of the proteasome established in this study, rather than the well-known canonical resting state ($20S_{resting}$) could be critical to future inhibitor development efforts that focus on allosteric rather than orthosteric sites of *Mtb* 20S CP. Aspects of the mechanism described here appear to be conserved in other structurally homologous systems. For example, allosteric activation of HslV protease, the *E. coli* homologue to the proteasome and the ancestor of archaebacterial and eukaryotic 20S proteasomes[57], involves the rotation of switch helices to correctly position the active site (Fig. 7), similar to what Losick and colleagues have proposed for the protein phosphatase 2C superfamily[42]. Interestingly, the binding of RPs or substitutions at the α-β interface of the archaeal *T. acidophilum* 20S CP modulate substrate specificity and alter the cleavage products[37]. These likely result from changes in the topology of the substrate binding pocket of *T. acidophilum* 20S CP due to the movement of the switch helices. Therefore, the mechanism described here may also apply to other structurally homologous systems, though, this needs to be experimentally tested. While our findings are based on in vitro experiments, they provide a foundational understanding of potential allosteric mechanisms that could operate under physiological conditions. Our work brings the mechanistic understanding of the *Mtb* proteasome system closer to the well-established archaeal and eukaryotic proteasomes and other structurally homologous systems.

## Methods

### Plasmids and constructs

Codon-optimized genes encoding the α- (Uniprot: P9WHU1, prcA, Rv2109c) and β-subunit lacking the propeptide (Uniprot: P9WHT9, prcB, Rv2110c) of *Mycobacterium tuberculosis* proteasome core particle (CP) were synthesized and inserted into separate pET24a vectors (Novagen, Madison, WI, USA) at the NcoI and BamHI restriction sites. For the α-subunit constructs, we incorporated a cleavable N-terminal $His_6$-SUMO tag. The β-subunit constructs were designed with a C-terminal TEV-SUMO-$His_6$ tag to enhance protein solubility during expression and to facilitate the subsequent removal of the tag after purification. Point mutations or deletions were introduced either using Quikchange site-directed mutagenesis or the corresponding genes were synthesized.

### Protein expression, and purification

Three distinct forms of the 20S proteasome CP were expressed, purified, and assembled: $20S_{WT}$, $20S_{\alpha\Delta2-6}$, and $20S_{\beta T1A}$. Unlike previous work that involved the co-expression of the α- and β-subunits[6,13], we designed a set of constructs along with a purification strategy to express and purify the α- and β-subunits separately. Following purification, these separately-expressed subunits were assembled to form the functional 20S CP. Plasmids were transformed individually into chemically-competent T7 Express *Escherichia coli* BL21(DE3) cells using the heat-shock method and were grown in lysogeny broth (LB) media containing 30 μg/mL kanamycin at 37 °C. When cultures reached an $OD_{600}$ of 0.6, the cells were induced with 1 mM isopropyl β-d-1-thiogalactopyranoside (IPTG) and protein expression was allowed to proceed for 18 hours at 16 °C. Cells were harvested by centrifugation at $4,000 \times g$ for 15 min at 4 °C and resuspended in lysis buffer (300 mM NaCl, 20 mM imidazole and 50 mM Tris-HCl, pH 7.0). The cells were lysed using an Emulsiflex-C3 high-pressure homogenizer (Avestin Inc., Ottawa, Canada) and the resulting homogenate was clarified by centrifugation at $23,700 \times g$ for 50 min at 4 °C. The supernatant was then passed over a $Ni^{2+}$-charged chelating FastFlow™ Sepharose column, which was subsequently washed with 40 mL of lysis buffer, and 30 mL of lysis buffer supplemented with 50 mM imidazole before finally eluting the protein using lysis buffer supplemented with 500 mM imidazole. The α- and β-subunits were subsequently combined and dialyzed against 300 mM NaCl, 1 mM DTT, 50 mM Tris-HCl, pH 7.0 for

18 h at 4 °C in the presence of TEV and Ulp1 proteases to remove the affinity tags and to promote the assembly of the 20S CP complex. The affinity tags and other impurities were then removed by passing the mixture over the IMAC column for a second time. The resulting flow-through was concentrated using a 10 kDa MWCO Amicon Ultra-15 centrifugal filter (Millipore). The fully formed 20S CP was separated from unassembled α- and β-subunits using size-exclusion chromatography (SEC) in 100 mM NaCl and 50 mM $NaH_2PO_4$, pH 7.4 as the running buffer. The 20S CP assembly yield was the highest for $20S_{WT}$, followed by $20S_{\beta T1A}$, $20S_{\alpha\Delta2-6}$. The αΔ2-6, βT1A construct also assembled into mature 20S particles, albeit with substantially lower yield. SDS-PAGE was used at each step to assess protein expression and purity. The final 20S CP concentration was determined spectrophotometrically with guanidinium chloride-denatured protein using extinction coefficients of 16,390 for the $\alpha_{WT}$-subunit, 14,900 for the $\alpha_{OG}$-subunit, and 20,400 $M^{-1}cm^{-1}$ for the β-subunits, obtained from Expasy ProtParam web-based tool (https://web.expasy.org/protparam/).

A codon-optimized gene encoding Mpa of *Mycobacterium tuberculosis* was cloned into a pET28 vector including a cleavable N-terminal $His_6$-SUMO tag. The plasmid was transformed into chemically-competent T7 Express *Escherichia coli* BL21(DE3) cells using the heat-shock method and were grown in Terrific Broth (TB) media containing 30 μg/mL kanamycin at 37 °C. When cultures reached an $OD_{600}$ of 0.6, the cells were induced with 1 mM isopropyl β-d-1-thiogalactopyranoside (IPTG) and protein expression was allowed to proceed for 4 hours at 37 °C. Cells were harvested by centrifugation at 4000 × *g* for 15 min at 4 °C and resuspended in lysis buffer (6 M GdnHCl, 20 mM imidazole, 50 mM Tris-HCl pH 8.0). The cells were lysed on ice via sonication for 5 cycles, where each cycle consisted of a 10 second pulse and 20 second incubation time. The resulting homogenate was clarified by centrifugation at 23,700 × *g* for 30 min at 4 °C. The supernatant was then passed over a $Ni^{2+}$-charged chelating FastFlow™ Sepharose column, which was subsequently washed with 40 mL lysis buffer, then 40 mL lysis buffer supplemented to 50 mM imidazole. An on-column refolding procedure was used using a series of washes of refolding buffer (150 mM NaCl, 20 mM imidazole, 50 mM Tris-HCl pH 8.0) with decreasing GdnHCl concentrations. Each wash used 50 mL of buffer and contained 4, 2 and 1 M GdnHCl, respectively. The column was then washed with 150 mL of non-denaturing wash (300 mM NaCl, 20 mM imidazole, 50 mM Tris-HCl pH 8.0) prior to elution in 50 mL elution buffer (500 mM NaCl, 500 mM imidazole, 50 mM Tris-HCl pH 8.0). The sample was then dialyzed against 500 mM NaCl, 1 mM DTT, 50 mM Tris-HCl, pH 8.0 for 18 h at 4 °C in the presence of Ulp1 protease to remove the affinity tag. The affinity tags and other impurities were then removed by passing the mixture over the IMAC column for a second time. The resulting flow-through was concentrated using a 30 kDa MWCO Amicon Ultra-15 centrifugal filter (Millipore) prior to SEC. The sample was separated using a Superose6 10-30 GL column using 500 mM NaCl, 1 mM DTT, 50 mM $NaH_2PO_4$ pH 7.5 solution as running buffer. SDS-PAGE was used at each step to assess protein expression and purity. The sample was further concentrated using a 30 kDa MWCO Amicon Ultra-15 centrifugal filter (Millipore), and the concentration was determined spectrophotometrically with GdnHCl-denatured protein using an extinction coefficient of 31,985 $M^{-1}cm^{-1}$ obtained from Expasy ProtParam web-based tool (https://web.expasy.org/protparam/).

A codon-optimized gene encoding dihydrofolate reductase (DHFR) from *E. coli* with an N-terminal *Mtb* Pup-tag was cloned into the pET-28 vector. A cleavable N-terminal $His_6$-SUMO tag was included as an affinity handle. This construct generates a linear Pup-DHFR fusion protein and has been used previously in structural and biochemical studies of Mpa:20S[19,41]. The plasmid was transformed into chemically-competent T7 Express *Escherichia coli* BL21(DE3) cells using the heat-shock method and were grown in LB media containing 30 μg/mL

kanamycin at 37 °C. When cultures reached an $OD_{600}$ of 0.6, the cells were induced with 0.2 mM isopropyl β-d-1-thiogalactopyranoside (IPTG) and protein expression was allowed to proceed for 18 hours at 16 °C. Cells were harvested by centrifugation at 4000 × *g* for 15 min at 4 °C and resuspended in lysis buffer (500 mM NaCl, 20 mM imidazole, 50 mM Tris-HCl pH 8.0). The cells were lysed on ice via sonication for 5 cycles, where each cycle consisted of a 10 second pulse and 20 second incubation time. The resulting homogenate was clarified by centrifugation at 23,700 × *g* for 30 min at 4 °C. The supernatant was then passed over a $Ni^{2+}$-charged chelating FastFlow™ Sepharose column, which was subsequently washed with 40 mL wash buffer (300 mM NaCl, 20 mM imidazole, 50 mM Tris-HCl, pH 8.0) then 40 mL wash buffer supplemented with 50 mM imidazole. The sample was eluted from the column using 50 mL 500 mM NaCl, 500 mM imidazole, 50 mM Tris-HCl, pH 8.0. Next, the sample was dialyzed against 500 mM NaCl, 1 mM DTT, 50 mM Tris-HCl, pH 8.0 for 18 h at 4 °C in the presence of Ulp1 protease to remove the affinity tag. The affinity tags and other impurities were then removed by passing the mixture over the IMAC column for a second time. The resulting flow-through was concentrated using a 10 kDa MWCO Amicon Ultra-15 centrifugal filter (Millipore) prior to SEC. The sample was separated using a S200 10-30 GL column using 500 mM NaCl, 1 mM DTT, 50 mM $NaH_2PO4$, pH 8.0 solution as running buffer. SDS-PAGE was used at each step to assess protein expression and purity. Sample concentration was determined spectrophotometrically with GdnHCl-denatured protein using an extinction coefficient of 34,950 $M^{-1}cm^{-1}$ obtained from Expasy ProtParam web-based tool (https://web.expasy.org/protparam/). Lastly, samples were spiked to 20% glycerol before freezing with liquid $N_2$ and storing at −80 °C.

## SEC-MALS
The oligomeric state of each 20S CP construct was probed using an OMNISEC multi-detector SEC system (Malvern Panalytical, United Kingdom) fitted with an OMNISEC RESOLVE and OMNISEC REVEAL modules. Both the autosampler and column oven were set to 20 °C. Injections of 100 μL with sample concentrations of 2 mg/mL were loaded onto a P3000 Protein SEC column (300×8 mm, Malvern Panalytical) equilibrated in 150 mM NaCl, 50 mM Tris-HCl, pH 7.5 at a flow rate of 1 mL/min. Molecular weight was calculated using light scattering detectors at 90° (right angle light scattering) and 7° (low-angle light scattering). BSA was used as a standard.

## Intact MS
Reverse phase liquid chromatography (LC) on a BEH C4 (1.7 μM, 2.1 ×150 mm) ACUITY column (Waters) connected to a Waters I-class ultra-performance liquid chromatography (UPLC) system was used to separate 1.5 – 2 pmol of each respective 20S variant (WT, OG or T1A). Proteins were separated on a 13-minute $H_2O$: acetonitrile (ACN) gradient at a flow rate of 0.2 mL/min and a column temperature of 40 °C. Both mobile phases were acidified using 0.1% [v/v] formic acid (FA). The ACN content was increased linearly from 2–85 % over 7 minutes, prior to a single sawtooth gradient where ACN was ramped from 2–85 % over 3 minutes before a final flush of 2 % ACN to equilibrate for the next injection. The LC eluent was directed to a Synapt G2-Si quadrupole time-of-flight (Q-TOF) mass spectrometer fit with a standard dual emitter electrospray ionization (ESI) source. For all the measurements, the instrument was operated in positive-ion mode with the capillary voltage set to +3 kV. The TOF mass analyzer was set to resolution mode, and surveyed precursor ions from 50–2,000 Th using 0.4 second scans. External calibration was performed with the "ESI-L low concentration tune mix" (G1969-8500–Agilent Technologies, CA). The instrument was also dynamically calibrated via infusing 2 μM Leucine Enkephalin (peptide sequence YGGFL; 1+ m/z 556.2771) dissolved in 50% [v/v] ACN and 0.1% [v/v] FA at 10 μL min$^{-1}$ from the lock mass sprayer, which was sampled every 20 seconds.

Data were analyzed using MassLynx (Waters) software. Spectra were summed over the two peaks of the chromatogram, which correspond to the elution profile of the respective α- and β-subunits. This corresponded to a 1.5-minute window of the chromatogram. A m/z range of 700–2000 Th was then used to deconvolute using the MaxEnt1 function, which used an output range of 23–30 kDa with a resolution of 1 Da/channel. A maximum number of 25 iterations was set prior to converging.

## Functional characterization

The peptidase activity of the 20S CP was measured using a pair of substrates: the tripeptide benzyloxycarbonyl-Val-Leu-Arg-7-amino-4-methylcoumarin (Z-VLR-AMC, Genscript) and a 11-residue oligopeptide conjugated to 7-methoxycoumarin-4-acetic acid referred to as LF2[14] (7-methoxycoumarin4-acetic acid (MCA)-Lys-Lys-Val-Ala-Pro-Tyr-Pro-Met-Glu-(dinitrophenyl)diaminopropionyl-NH2, Genscript). Substrate stocks were prepared in MQ water with 10% DMSO (v/v) at 10-fold the intended final concentration. Each of the 20S proteasome constructs at a concentration of 21.4 nM were incubated with 0–175 μM of Z-VLR-AMC and 0–30 μM of LF2. At [Z-VLR-AMC] exceeding 175 μM, we observed an inhibition of catalytic activity, which is most likely due to substrate inhibition. Consequently, initial velocities corresponding to [Z-VLR-AMC] concentrations above this threshold were excluded from data fitting. All reactions were performed at 25 °C in 20 mM NaCl, 10 mM NaH$_2$PO$_4$, pH 7.4 and 1% DMSO (v/v). Reactions were initiated by addition of protein and monitored using a Cary Eclipse fluorometer (Agilent Technologies, Mississauga, Canada). Cleavage of Z-VLR-AMC was monitored using 380 nm and 450 nm excitation and emission wavelengths, respectively, with a bandpass of 5 nm. Degradation of LF2 was monitored at an excitation wavelength of 340 nm and emission wavelength of 405 nm with a bandpass of 5 nm. LF2 was titrated from 0–50 μM against 1 μM MCA to correct for the impact of LF2 on MCA fluorescence[58]. The resulting data were fit to a second-order polynomial, and the correction was applied to the measured velocities. Standard curves of AMC or MCA were used to convert relative fluorescence values into the concentration of product formed. Initial velocities were extracted from the first 30 seconds of each reaction using Python scripts written in-house.

Inhibition of the peptidase activity of the 20S WT CP against Z-VLR-AMC was tested using the peptidyl boron-based inhibitor, ixazomib (Selleck Chemicals LLC). Measurements were taken as described above, with a final Z-VLR-AMC concentration of 150 μM and inhibitor concentrations ranging from 0.35–10 μM. Both enzyme and substrate stocks were prepared with the appropriate concentration of inhibitor and 1% DMSO (v/v), for a final concentration of 1% DMSO (v/v) in the reaction. Relative activity was then calculated against the uninhibited reaction.

All functional assays were carried out in technical triplicates. Data for both the functional characterization and inhibition studies were fit to the Hill model (see below) and visualized using in-house scripts written in Python v3.8.

$$V_o = \frac{V_{max}\,[S]^n}{[S]^n + K_{0.5}{}^n}$$

$$Normalized\,V_o([I]) = \frac{V_o([I]) - V_o([I_{max}])}{V_o([I_{min}]) - V_o([I_{max}])} = \frac{1}{1 + \left(\frac{[I]}{IC_{50}}\right)^n}$$

We estimated uncertainties in the fitted $V_{max}$, $n$, $K_{0.5}$, and apparent IC$_{50}$ parameters using a Monte Carlo approach[59], wherein we introduced random errors, determined by the root-mean-square deviation between experimental points and those from the optimal-fit model, into the best-fit model to generate 100,000 synthetic datasets. These datasets were then fit as per the experimental data. The values obtained from the Monte Carlo iterations were converted to a histogram, which was subsequently fit to a normal distribution function to yield mean expectation values and standard deviations ($\sigma$). Log-transformation was used to better estimate the uncertainty of fitted parameters with asymmetric error distributions. Final uncertainties are presented as $2\sigma$ in the derived values, representing a 95% confidence interval. All relevant scripts are available upon request.

The ability of the Mpa:20S complex to degrade the model substrate, Pup-EcDHFR, as monitored through SDS-PAGE was tested against three 20S variants: WT, OG and βV53Q. Reactions containing 0.1 μM 20S CP, 5 μM Pup-EcDHFR, and 0.2 μM Mpa hexamer were carried out at 37 °C in reaction buffer (100 NaCl, 1 mM DTT, 25 mM MgCl$_2$, 50 mM NaH$_2$PO$_4$ pH 7.5) over 25 hours. The reactions were initiated through the addition of 7.5 mM ATP. Time points were taken at 0, 2.5, 5, 8 and 25 hours which were quenched through the addition of Laemmli buffer. Protein bands were separated using SDS-PAGE on a 20% Tris-glycine gel with Blue Elf ladder (Froggabio) used as reference.

## Cryo-electron microscopy sample preparation and data collection

All 20S CP variants were concentrated to 7 μM, equivalent to 98 μM of each of the α- and β-subunits. For the ixazomib-bound state, the inhibitor was added to a final concentration of 110 μM in a final DMSO concentration of 0.5% (v/v). Each sample was applied to a holey carbon grid (C-flat CF-2/1-3Cu-T) that had been previously glow-discharged in air at 10 mA for 15 sec. Sample vitrification was performed using a Vitrobot Mark IV (Thermo Fisher Scientific) at 4 °C and 100% humidity. Each sample was incubated on the grid for 3 sec and the grid was blotted for 5 sec with a blot force of 1 before plunging in liquid ethane. Datasets were collected using the SerialEM software(v4.1.0β)[60] on a Titan Krios microscope (Thermo Fisher Scientific) at the Facility for Electron Microscopy Research at McGill University. Movies were recorded on a Gatan K3 direct electron detector equipped with a Quantum LS imaging filter. The total dose used for each movie was 50 e$^-$/Å$^2$, equally spread over 30 frames. The datasets were collected at a magnification of ×105,000, yielding images with a calibrated pixel size of 0.855 Å. The nominal defocus range used during data collection was between −1.25 μm and −2.75 μm. Data collection parameters for all the datasets are summarized in Supplementary Data 1.

## Cryo-electron microscopy data processing

All cryo-EM data processing was performed using cryoSPARC v4[61]. The movies were corrected for beam-induced motion using patch motion correction, and the CTF parameters were estimated using patch CTF estimation. Micrographs with CTF were fit to a resolution worse than 10 Å, and those with unusually high full-frame motion were discarded at this stage. The remaining micrographs were used for particle picking using blob picker with a minimum and maximum diameter of 100 Å and 250 Å, respectively. These particles were subjected to two rounds of 2D classification, and the selected classes were used as templates for template-based particle picking. The template-based picked particles were subjected to two rounds of 2D classification for particle curation. The particles from the best classes were then used for ab-initio reconstruction and heterogeneous refinement with 5 classes. The classes displaying the 20S CP characteristic structural features were selected and used for high-resolution homogenous refinement. These particles were subjected to reference-based motion correction in cryoSPARC to correct for beam-induced motion at the per-particle level. These motion-corrected particles were subsequently used to obtain the final high-resolution map with D7 symmetry enforced, combined with global and local CTF refinement[62]. Local resolution was estimated for all the maps using cryoSPARC v4. ChimeraX (v1.8)[63] was used for visualizing the maps and figure preparation. Data processing details for all the datasets are summarized in Supplementary Figs. 8–11.

We used cryoSPARC's 3D variability analysis (3DVA)[52] module to probe conformational heterogeneity in our datasets. The particles corresponding to the final refined maps were symmetry expanded to D7 symmetry, and the resulting particles were used as input for the 3DVA with six variability components and a low-pass filter resolution of 3.5 Å. In the case of 20S$_{\beta T1A}$, the third principal component was subjected to 3DVA intermediate display with a window factor of 0 to obtain 20 non-overlapping classes of particles. These sorted symmetry expanded particles were then used for local refinement to obtain high-resolution maps of the intermediate states. The maps corresponding to the two endpoint classes (i.e., frames 1 and 20) were used for atomic modelling.

## Model building and refinement

To build the molecular model for wild-type 20S CP, *Mycobacterium tuberculosis* 20S CP (PDB 8D6V)[41] was used as an initial template. For all the other datasets, the refined wild-type model was used as the template. The template model was initially docked onto the maps using ChimeraX[63]. For the ixazomib-bound dataset, the restraints for the threonine hydroxyl-boron bond were generated using JLigand (v1.0.40)[64]. Model building was performed by an iterative cycle of manual building in Coot (v0.9.8.92)[65] and real space refinement in PHENIX[66], which significantly improved the model's geometry and fit in the map. Model validation was also performed on PHENIX(v1.21rc1_5109)[66], and the model statistics are summarized in Supplementary Data 1.

## HDX-MS

Continuous labelling, bottom-up hydrogen-deuterium exchange mass spectrometry was performed on four states: unbound (apo) 20S$_{WT}$, ixazomib-bound 20S$_{WT}$, 20S$_{OG}$, and 20S$_{\beta T1A}$ (Supplementary Table 1). All solutions used in these experiments contained 100 mM NaCl and 50 mM NaH$_2$PO$_4$, pH 7.4. D$_2$O-based solutions were adjusted to pD 7.4 using the standard electrode correction procedure[67,68]. An initial equilibration step was carried out for each experiment, in which protein stocks were diluted to 2 μM into a H$_2$O-based buffer for a minimum of 30 min. 10 μM ixazomib was present in all solutions for the ixazomib-bound state. HDX was initiated by a ten-fold dilution into D$_2$O-based buffer that contained identical additives to that of the equilibrated mixture. HDX was conducted at room temperature with labelling times over 0.167–1440 min. Reactions were quenched by acidification to pH$_{read}$ 2.5 by mixing sample aliquots 1:1 (v/v) with 3 M guanidium chloride, 250 mM NaH$_2$PO$_4$, pH 1.52, and 3 mM n-dodecylphosphocholine[69] (final concentration of 1.5 M GdnCl, 125 mM NaH$_2$PO$_4$, and 1.5 mM n-dodecylphosphocholine) and flash frozen in liquid N$_2$. Samples were stored at −80 °C prior to analysis.

Pulsed bottom-up hydrogen-deuterium exchange mass spectrometry was performed on four variants: 20S$_{WT}$, 20S$_{\alpha Q98K}$, 20S$_{\beta A92-94G}$, 20S$_{\beta A100S}$ (Supplementary Table 2). All solutions used in these experiments contained 100 mM NaCl and 50 mM NaH$_2$PO$_4$, pH 7.4. D$_2$O-based solutions were adjusted to pD 7.4 using the standard electrode correction procedure[67,68]. An initial equilibration step was carried out for each experiment, in which protein stocks were diluted to 2 μM into a H$_2$O-based buffer for a minimum of 30 min. HDX was initiated by a ten-fold dilution into D$_2$O-based buffer that contained identical additives to that of the equilibrated mixture. HDX was conducted at room temperature over 10 seconds before being quenched by acidification to pH$_{read}$ 2.5 by mixing sample aliquots 1:1 (v/v) with 3 M guanidium chloride, 250 mM NaH$_2$PO$_4$, pH 1.52, and 3 mM n-dodecylphosphocholine[69] (final concentration of 1.5 M GdnCl, 125 mM NaH$_2$PO$_4$, and 1.5 mM n-dodecylphosphocholine) and flash frozen in liquid N$_2$. Samples were stored at -80 °C prior to analysis.

Liquid handling and reverse-phase separation was performed using a M-Class nanoAcquity UPLC system equipped with HDX technology (Waters, Milford, MA, USA). 5 pmol of the sample was digested at 15 °C using a nepenthesin-2 column (Affipro, AP-PC-004, 1 mm × 20 mm, 16.2 μL). The resulting peptides were trapped for 3 min on a BEH C18 (1.7 μm, 2.1 mm × 5 mm; Part#: 186003975, Waters) column at a flow rate of 100 μL min⁻¹. Peptide separation was achieved on an HSS T3 (1.8 μm, 1.0 × 50 mm; Part#: 186003535, Waters) column using a linear, 8-minute acetonitrile: H$_2$O gradient acidified with 0.1% formic acid (acetonitrile ramped from 5–35%) at 0 °C and at a flowrate of 100 μL min⁻¹. Between each injection, the sample loop and the protease column were flushed using 1.5 M guanidine hydrochloride, 4% (v/v) acetonitrile, 0.8% (v/v) formic acid, and 1.5 mM n-dodecylphosphocholine. This routine proved effective in reducing column carry-over to undetectable levels for our proteasome samples, as judged by blank injection using the same inlet and MS methods.

The ULPC outflow was directed to a quadrupole ion mobility time-of-flight (Q-TOF) Synapt G2-Si mass spectrometer (Waters) fitted with a standard electrospray source operated in positive-ion mode with a capillary voltage of +3 kV. Online calibration of the instrument was achieved by infusing LeuEnk solution (1+, 556.2771 Th) from the LockSpray capillary every 20 seconds at a flow rate of 10 μL min⁻¹. Drift time-aligned MS$^E$ data-independent acquisition was employed for peptide mapping, as described previously[70]. Data were acquired over the 50–2000 *m/z* range with a scan time of 0.4 sec. Fragmentation was induced by linearly increasing the transfer collision energy in alternating scans over 20–40 V. Ion mobility separation was controlled manually as described previously[70]. The quadrupole was manually set to dwell at 300 *m/z* to exclude smaller ions. The TOF mass analyzer was operated in resolution mode. We performed additional peptide mapping experiments for 20S CP samples that included mutations as needed. This was not necessary for ixazomib-bound samples as ixazomib forms a reversible bond with the β-subunit and rapidly dissociates upon denaturation and leaves a chemically unmodified enzyme.

Undeuterated (reference) and deuterated samples were all repeated in technical triplicates. Reference data collected from MS$^E$ experiments provided sequence identification using Waters ProteinLynx Global Server (v3.0.3) searched against a database containing the plasmid sequences of both the α- and β-subunits. This database also included variations corresponding to any mutations present in each specific subunit. Peptide filtering parameters were taken from Sørensen et al.[71]. The spectra corresponding to the peptides that satisfied our stringent filtering criteria were manually inspected, and only those with high-quality spectra and high signal-to-noise ratios were retained for further analysis. The level of deuterium uptake for peptides of interest at time *t* was referenced to an undeuterated control. The undeuterated control underwent identical preparation steps, with the exception that all solutions were H$_2$O-based. The fully deuterated control was prepared as described by Peterle et al.[72]. Back-exchange was calculated using the maximally deuterated controls[72]. It should be noted that deletions or mutations were not found to significantly change the predicted $k_{ch}$[73]. Data analysis was carried out using DynamX v3.0 (Waters). We used a hybrid significance test model that selected peptides with deuterium uptake differences greater than ± 0.5 Da and that also passed Welch's *t*-test with α < 0.01, to ascertain significant variations in deuterium uptake[74]. The heat maps were generated using HDgraphiX[75].

## Reporting summary

Further information on research design is available in the Nature Portfolio Reporting Summary linked to this article.

## Data availability

The electron microscopy density maps generated in this study have been deposited in the Electron Microscopy Databank under accession codes: EMD-45494, EMD-45495, EMD-45496, EMD-45498, EMD-45499,

EMD-45501, EMD-45532, EMD-45534, EMD-45535, EMD-45537, EMD-45538, EMD-45539, EMD-45540, EMD-45541, EMD-45542, EMD-45547, EMD-45552, EMD-45553, EMD-45556, EMD-45558, EMD-45559, EMD-45560, EMD-45561, EMD-45562. Atomic models generated in this study have been deposited in the Protein Databank under accession codes: PDB ID 9CE5, PDB ID 9CE7, PDB ID 9CEB, PDB ID 9CE8, PDB ID 9CEE, PDB ID 9CEG. Previously published *Mtb* 20S CP structures were retrieved from the Protein Databank under accession codes PDB ID 3MI0 and 3MKA respectively]. The mass spectrometry data available from the MassIVE database as entry MSV000095758. The DNA sequences for the plasmids used are deposited at the NCBI database under entries: PQ217960, PQ217961, PQ217962, PQ217963. Source data are provided with this paper.

## Code availability

The code used for the analysis of biochemical assays has been deposited in the Zenodo database under https://doi.org/10.5281/zenodo.14990241.

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

## Acknowledgements

M.T. acknowledges support from a Natural Sciences and Engineering Research Council of Canada Postgraduate Doctoral Scholarship. Financial support was provided by a Canadian Institutes of Health Research Project Grant PJT451412 to S.V., and a Natural Sciences and Engineering Research Council of Canada Discovery Grant (RGPIN/03031-2022) to N.Z. MS data were recorded at the Mass Spectrometry

Facility of the Advanced Analysis Centre, University of Guelph. We thank Dr. Dyanne Brewer (University of Guelph) for assistance with MS measurements. We thank Prof. M. Strauss, K. Basu, and K. Sears at the Facility for Electron Microscopy Research (FEMR) at the McGill University for their assistance with microscope operation, data collection, and computational support. FEMR is supported by the Canadian Foundation for Innovation, Quebec Government, and McGill University. N.Z is a member of the Centre de recherche en biologie structurale, funded by Fonds de Recherche du Québec (Health Sector) Research Centres Grant no. 288558. We thank Profs. Lewis Kay (University of Toronto), Matthew Kimber (University of Guelph), and Robert Harkness (University of Guelph) for guidance and helpful discussions.

## Author contributions

M.T., E.R. and S.V. initiated the project; M.T., A.B.U., E.R., A.V., N.Z. and S.V. designed research; M.T., A.B.U., E.R., A.V., N.Z. and S.V. performed research; M.T., E.R., A.V. and S.V. contributed new reagents/analytic tools; M.T., A.B.U., N.Z. and S.V. analyzed data; M.T., A.B.U., N.Z. and S.V. wrote the paper; N.Z. supervised the electron cryomicroscopy studies; and S.V. supervised the mass spectrometry and biochemical studies.

## Competing interests

The authors declare no competing interests.
