## [Transparent Peer Review file · Nature Communications]

Structural basis for allosteric modulation of *M. tuberculosis* proteasome core particle

Corresponding Author: Professor Siavash Vahidi

Version 0:

Reviewer comments:

Reviewer #1

(Remarks to the Author)

Turner et al. presents in this manuscript a structural study of Mtb 20S proteasome core particle by cryo-EM and HDX-MS. The major contribution of this work is that the authors were able to reconstruct a so-called “inactive” state of the Mtb 20S complex, with a few detailed structure features distinguishing this state from those previously reported. This work potentially provides new insights into the allosteric regulation of the Mtb 20S and paves the way for structure-based therapeutic discovery against Mtb. This work might be considered for publication if the authors can address the following issues.

- (1) The naming of so-called “inactive” state of their newly observed 20S conformation may not appropriate. The authors appear to insufficiently understand the current stance and literature of the proteasome field and over-simplifying the process of proteasome activation into a two-state “switch”-like process. Almost all major literature regarding proteasome activation were missing and ignored in the references section. The 20S core particles have been resolved in “inactive” states that were inhibited by small molecules at the active sites, although structural insights into the catalytically active state were gleaned through these inactive inhibited 20S structures. More importantly, the 20S CP activation is now much better understood as a multi-step process, which involves pro-peptide cleavage at the active site, assembly of RP onto CP for priming but not immediately activation, and then substrate engagement, followed by deubiquitylation and large conformational transition of RP before the proteasome can be fully activated (the authors are referred to the recent literature of substrate-engaged human and yeast proteasome structures in Nature and Science). In eukaryotes, this process is further orchestrated by proteasome-associated proteins, like USP14/Ubp6 and UBE3C/Hul5. In many published works, even the 26S proteasome in a ground, resting state is considered “inactive”. Therefore, the claim that the “inactive” state of Mtb 20S CP in this work is the first inactive state of 20S ever discovered does not make much sense and represents a substantial ignorance and misunderstanding to the existing literature and knowledge accumulated in the proteasome field. From what the authors described, the so-called “inactive” state might be more like a “self-inhibited” or “auto-inhibited” state, where the switch helices block the access of substrate.
- (2) Following the above major concern, the authors should also avoid naming the state of 20S as “active” state. It may be better called “canonical resting state”.
- (3) On line 421-423, the sentence reads confusing or inappropriate in that there is completely no data shown in this work that the so-called 20S^{off} conformation and associated mechanism is conserved across the three domains of life. Extrapolating structural observation on Mtb to a broader context, in this case, should be very careful and should adhere to the data and evidence. The heteroheptamers of the CP in eukaryotes are considerably more complex and well differentiated as compared to the homoheptamer assembly of Mtb 20S. This reviewer cannot believe that anything observed on Mtb 20S must be conserved in eukaryotes unless there is solid evidence.
- (4) The authors described that they used 3DVA to reveal the conformational variation. Can they show the 3DVA results in a statistical distribution of particles mapping to 2 or 3 dimensions of principal components in one of the Extended Data Figure? Can they show their classification of the 20 frames along one of the principal components they choose to visualize the conformational changes?
- (5) Although the authors did in vitro test of their 20S^{off} conformation model, there is no data to clarify if the 20S^{off} conformer is physiologically relevant or did exist in cells. Even indirect evidence should be helpful in this case.
- (6) Alternatively, the authors need to conduct validation experiments to prove that the 20S^{off} conformer is not caused by inappropriate biochemical treatment of proteasome samples and is not an artificially triggered, destabilized proteasome conformation that mostly does not exist along the functional pathway of the proteasome.
- (7) As mentioned above, allosteric regulation of the proteasome CP is a lot more complicated than just regulating the 20S

core particle's proteolytic active site. This paper only addresses a very limited aspect of the allosteric regulation of the 20S CP in Mtb, and there is no evidence of 20Soff conformers will ever be observed in any species like yeast or human cells. Therefore, it is highly recommended that the authors revised the title of the paper to be specific and adhere to their core finding rather than generally claiming that they "finally" discover the structural basis for allosteric regulation of the proteasome CP "once for all", which we know it does not. Something like "Structural basis for auto-inhibition of active sites and its allosteric regulation in Mtb 20S proteasome core particle" would better serve the scholarly communication in this case.

Reviewer #2

(Remarks to the Author)

In this manuscript, the authors investigate the structure of the *Mycobacterium tuberculosis* 20S proteasomal core particle in the absence of a regulatory particle using single-particle cryo-EM and hydrogen-deuterium (H/D) exchange experiments. Proteasomes in all domains of life are gated against random substrate access to the proteolytic chamber. Wild-type 20S particles degrade only small peptides that can diffuse into the proteolytic chamber without gate-opening, whereas larger peptides and proteins require regulatory complexes that associate with the 20S ends to facilitate gate opening and substrate entry.

Turner et al. determined four structures: wild-type 20S (20Swt), a mutant where the active site threonine is substituted with alanine (20S T1A), an open-gate variant with the first seven residues of the alpha subunit deleted (20S OG), and 20S bound to a peptidyl boronate inhibitor (20S ixazomib). The authors propose that the structure of the mutant 20S proteasome, 20S T1A, represents an "inactive" or "20S OFF" state that prevents substrate binding based on three observations: the WT 20S and the 20S OG have mildly different affinities and Hill coefficients for peptide substrates, there is a small shift of a helix in the vicinity of the active site observed in the T1A mutant structure, mutations in the alpha subunit distant to the active site lead to a modest reduction in the rate of the reaction and the affinity for the substrate.

Although the experimental data is technically sound, the manuscript suffers from overinterpretation, and the conclusions are not justified by the experimental evidence. Unless the authors can provide convincing experimental evidence that the observed conformation is indeed relevant for the activity of the wild type enzyme, the manuscript is not a strong candidate for further consideration.

Below are some specific points of concern:

1. The abstract claims that "Both eukaryotic and Mtb proteasome systems are allosterically regulated, yet the specific conformations involved have not been captured in high-resolution structures to date. Here we present the first structure of ... any 20S CP in an inactive state ... distinguished from the canonical active state... by the conformation of switch helices I and II. The rearrangement of these helices collapses the S1 pocket, effectively inhibiting substrate binding." These statements imply a regulatory mechanism that restricts substrate access, whereas such access is primarily controlled by the alpha-ring gating and interactions with regulatory particles. The 20S wt particle, in the absence of a regulatory complex, does not exhibit an "OFF" state. In fact, all previously solved structures of 20S wt, as well as the one in this study, are in the "ON" state. Furthermore, the purported "OFF" conformation does not prevent substrate binding; rather, it transitions to the "ON" state upon substrate binding (the authors describe a dynamic fluctuation between these two states). This structural plasticity is typical of enzymes during their reaction cycle, but does not constitute a regulatory "switch."
2. Previous high-resolution structures of 20S CPs, including the crystal structure of the 20S T1A mutant (Li et al., 2010), have consistently shown the conformation that the authors describe as "ON." How do the authors explain the divergence between their 20S T1A structure and the one published by Li et al.?
3. The authors claim that peptidyl boronates mimic the transition state of hydrolysis and were used in the hope of stabilizing "inactive" conformations. However, transition-state mimics should theoretically stabilize on-pathway conformations. Can the authors clarify this statement, particularly considering that their ixazomib-bound structure corresponds to the "ON" state, rather than an inactive state?
4. The authors do not observe an "OFF" state for 20S wt, nor do they identify changes in the alpha subunit conformations between the "ON" and "OFF" states. Yet, in lines 190-200, they argue that the decreased local resolution in the 20S wt alpha rings suggests interconverting conformations and, by extension, an allosteric network that spans to the alpha subunits. These conclusions are an overinterpretation and are not supported by the data. Furthermore, it is not a new concept that an allosteric network exists between the alpha and beta subunits, but the data provided here do not contribute new information about the nature of this allosteric network between alpha and beta subunits.
5. The H/D exchange experiments reveal changes in flexibility and local conformation between the 20S variants (wt, T1A, and ixazomib-bound), but they do not provide sufficient structural evidence for defining the allosteric network. Similarly, the cryo-EM structures show congruent alpha subunit conformations across all four variants. The experimental evidence does not support the conclusions.
6. Point mutations in the alpha subunits and beta subunits were generated to support the existence of an inactive state. The authors argue that the v_{max} of peptide cleavage at the beta subunits can be influenced by mutations in the alpha subunits.

However, the observed changes are very mild (v_{max} changes from 3 to 4 μM , with $K_{0.5}$ changes from 95 to 60 μM) and do not support a regulatory “activity switch”. The mutations introduced in the beta subunits (Y35F and V53Q) likely directly disturb active site residues, since Y35 H-bonds to Lys33 in the active site and to Val53. It is therefore not surprising to observe changes in catalytic parameters.

7. The main established role of the 20S proteasome in *M. tuberculosis* is the degradation of pupylated substrates by the Mpa/20S complex. The relevant question is therefore, how the postulated “activity switch” affects degradation of bona fide protein substrates in the context of the full degradation complex. The authors do not provide any data on this. The authors should test, how their point mutants behave in degradation of pupylated substrates by the Mpa/20S complex.

8. Lastly, the proposal of an “activity switch” implies that the 20S is in an inactive state and needs to be activated by a specific trigger. What is the mode of activation? Literature would suggest that it is the binding of regulatory particles. In that case, the wt 20S in isolation should remain inactive and should only be activated in presence of the regulator. This is however not what the authors observe, since their 20S wt exhibits an “ON” state and shows degradation activity towards the tripeptide substrate.

Reviewer #3

(Remarks to the Author)

In this article, Turner and coworkers decipher the structural bases responsible for the allosteric regulation of the Mtb 20S core proteasome:

- They solved four near-atomic resolution (2.5 – 2.7 Å) structures of Mtb 20S particles by cryo-EM: apo and ixazomib bound WT 20S, 20S β T1A and 20SOG variants.
- While the structure of WT bound to ixazomib shows a binding mode that is similar to that observed in human 20S, comparison of apo WT vs. 20S β T1A shows that the β T1A mutation is enough to slightly rotate the Switch Helix 1, which occludes the S1 pocket, impeding further binding of ixazomib. Importantly, this represents the first structure of an inactive form of the 20S proteasome.
- Substrate degradation comparison of WT vs β T1A vs OG conformers shows an increased cooperativity of OG vs. WT.
- The comparison by HDX-MS of the apoWT with the ixazomib-bound-WT and OG & T1A mutants highlighted some differences in dynamics that could not be observed by cryo-EM.
- The discovery of mixed EX1-EX2 regimes (in the ixazomib bound WT and T1A mutant), characteristic of slow interconversion changes is nicely confirmed and illustrated by the 3DVA approach. Such analysis of the T1A mutant shed light on the interconversion of the switch helices I and II between two conformers that were assigned as active (on) and inactive (off) states of the 20S, respectively.
- Site-directed mutagenesis rationalized based on HDX-MS, cryoEM results and previous results from the literature allowed to further describe the allosteric dialogue between the alpha and beta rings. A double mutant showed that the positive effects observed on the alpha and beta rings could be additive. Finally, the fact that the longer substrate could not be proteolyzed by any of these mutants, nicely shows that these reciprocal allosteric changes are uncoupled with the opening of the gate. Overall, this very well written and illustrated article confirms and further describes a general mechanism of proteasome bidirectional allostery observed throughout the evolutionary tree. Besides, the cryo-EM structures provided here, together with the HDX-MS analysis, provide striking evidence for a major role played by the switch helix 1 (beta subunit) in this allostery, conveying structural changes from the beta to the alpha ring, and reversely. The experiments were perfectly well conducted and the main results confirmed by multiple complementary approaches.

For all these reasons, this article deserves to be published in Nat Commun, with minor corrections, as detailed below:

- In the introduction, the authors mention an article from S. Gygi and coworker, suggesting the presence of long-range allosteric regulation in Mtb 20S proteasome. However, no mention is made in the introduction to similar results/observations made in 20S proteasome for other prokaryotes (Ruschak et al, PNAS 2012), yeast & mouse (Arciniega, 2014, PNAS) or human (Osmulski and Gaczynska 2002, Cell / Rego and da Fonseca Mol Cell 2019 / Lesne et al, 2020 Nat Commun). These articles, showing allosteric regulation following binding of regulatory particles or binding/modification of the catalytic residues, include results coming from AFM, NMR, CryoEM and HDX-MS experiments. Most of these articles are actually discussed in the discussion, but reference should also (or alternatively) be made in the introduction, to present the state of the art in terms of 20S allostery, as a preamble to this study.
- P4. L91: did the authors try to assemble double mutant 20S β T1A/OG?
- P5. L114: “The shift in the Hill coefficient and $K_{0.5}$ values between the 20SWT and 20SOG variants points to increased cooperative interactions when the gating residues are removed”. The reviewer is wondering if this increase in the Hill coefficient is really due to a better affinity of the substrate for the catalytic sites or to a better accessibility to these sites (higher diffusion) thanks to the removal of the gate? Is it really the allosteric cooperativity that is increased or rather the steric hindrance that is decreased?
- P5.L114: “Together, these results support the view that the 20S CP is an allosteric system which interconverts between multiple conformations that differ in their activity and affinity for the substrate.” At this stage (WT vs. OG), the referee is not sure that this assertion can be made, since the increase in activity could only reflect an increase in accessibility for the substrate, which is not necessarily due to allosteric changes. The subsequent T1A mutation is probably a better example of long-range allosteric change. This sentence could there be placed later on in the manuscript.
- P5.L125: Did the authors try to measure the affinity and Hill coefficient of ixazomib for the OG variant?
- P8.L213: The Nter of the β subunit is marked as covered in the peptide mapping (Extended Data Fig.6), even in the ixazomib bound WT. Since the binding is supposed to be covalent, the addition of the ligand will increase the MW of these Nter peptides: did you monitor the HX on these covalently modified peptides or on the portion of un-liganded Nter peptides?

Similarly, the T1A mutation will also modify the MW of these Nter peptides, meaning that this region cannot be differentially analyzed. Could you comment on this?

- P9.L218-227: While the protection effect induced by ixazomib on the beta subunit is very clear, the two following assertions concerning subunit alpha seem a bit weaker and should be tuned down. "Of note, ixazomib binding also induced changes in the conformational dynamics of the α -subunit." These are very slight changes compared to the 1TA or OG mutants. Then, concerning residues alpha1-12: "These measurements are supported by multiple partially overlapping peptides". In Extended Data 6, only the 1st timepoint of peptide 1-12 is shown as deprotected, but none of peptides 1-6, 1-14, 7-12, so what are these overlapping peptides? This is clear for the protection of the Swith Helix I, but not for the slight changes observed in subunit alpha.
- P9.L236-237: "Likewise, peptides at the α - β interface also demonstrated reduced deuterium uptake, consistent with the changes induced by ixazomib binding. However, the changes span an entire helix in the α -subunit (peptides containing α 75-90) rather than just those residues localized to the site closest to the β -ring." This sentence seems to suggest that the interface is more protected in the 1TA mutant vs. in the ixazomib sample, which is actually the contrary, since in both Extended Fig.6 and 7, the peptides from subunit beta protected in 1TA at the alpha/beta interface are less numerous (87-98) than in the ixazomib sample (85-91/87-91/87-98/99-104/99-110).
- P9.L241: "and the structural elements involved in RP binding" please highlight these residues in Extended Data Fig.6a.
- P10.L246-257: The results observed on the OG mutant are striking and make sense. However, the increased deuteration is explained here only as a propagation of conformational changes, whereas the physical removal of the gates probably eases the diffusion of deuterons inside the 20S cavities. Therefore, an increased accessibility to the (deuterated) solvent could also be partly responsible for this increased deuteration.
- P11.L259: While the Figures of the manuscript are generally very well designed, the Reviewer is wondering about the impact/efficiency of Fig.3a. The top part of the panel seems unnecessary (since the bottom part is enough and self-explanatory). However only the timepoint at 10sec is represented here. The Reviewer suggests to remove the top part of panel a and replace it by the heatmaps of Extended Data Fig6.
- P13.L294: "This motion is present in each 20S variant, except for the ixazomib-bound 20SWT variant, suggesting that inhibitor binding may limit the overall mobility of the 20S CP." Could this be related to the differences in local resolution (Extended Fig.5) that show indeed some "rigidification" of the ixazomib bound 20S, compared to the WT and OG mutants? However, the resolution is even better for the T1A mutant, but the motion described in the 3DVA analysis is nonetheless present... Could the author comment on that?
- P14.L335-371: The analysis of the different mutants corroborates the proposed allosteric mechanism. One might then wonder how would the dynamic "breathing" of these constructs be affected. Would it be exacerbated in the alphaK52F/betaV53Q double mutant and strongly reduced in the betaY35F mutant? Although such experiments probably go beyond the scope of this article, could the authors propose working hypotheses?
- P14.L335-371: Did you try to combine these mutations with the original OG and betaT1A variants and measure their activity?
- P17.L412-416: "Molecules that selectively bind to the inactive conformation (20SOFF) of the proteasome established in this study, rather than the well-known active form (20SON) could be critical to future inhibitor development efforts that focus on allosteric rather than orthosteric sites of the proteasome in bacterial and eukaryotic systems." Do you suggest as an SAR strategy to stabilize the inactive 20S form by filling up the S1 pocket or by inducing this conformational change of the Swith Helix I? Given the presence of three different catalytic subunits in eukaryotic 20S, do you think that it would be possible to target all three subunits concomitantly with the same drug, or that impairing just one would be enough?
- P18.L456: The MassIVE repository access number should be stated for proper reviewing.
- P26.L674: "Initial velocities were extracted from the first 30 seconds of each reaction using Python scripts written in-house". Did you check, or does your Python scripts take into account the linearity of the slope to measure the activity? Some lag in this linear region might appear, which may affect the calculated values.
- P30.L782: The authors did inject some GdHCl to clean the sample loop but it seems that they did not run a proper blank (same chromatographic run) between each injection, which is a good practice to avoid carry-over on the C18 column. This is particularly crucial when EX1 or mixed EX1-EX2 are observed, to avoid mis-interpretation of the dataset. The fact that most peptides do not show this mixed behavior is not enough, since not all peptides bind the column with the same affinity. Did the authors make sure beforehand that no carry-over was present in these conditions?

Minor comments:

P3.L57: In legend of Fig1; the authors refer to the PDB 3M10, whereas the latter corresponds to a substrate-free form of Arginine Kinase...In panel c, the authors refer to structures on the left (CP-WT) and on the right (CP-OG), however the structure on the right (cartoon) is part of panel d...Is this an error or are the authors talking about the structure from panel d? Please clarify

P4. L73: please remove "recently", since the ref is almost 10 years old.

Extended Data Fig. 4 is cited in the text before extended Data Fig. 3.

P13.L319: "Concurrently, residues 92-98 of switch helix II show increasing order, with their density becoming well-defined by frame 20" suggestion: "explaining the slower deuteration uptake of peptide 87-98 (Extended Data Fig.6).

P.28L744: "These sorted symmetry expanded particles were..."

Extended Data Figures:

P3.L846: please check ref. 3.

Version 1:

Reviewer comments:

Reviewer #1

(Remarks to the Author)

The authors have addressed all my questions.

Reviewer #2

(Remarks to the Author)

I appreciate the effort the authors have made in addressing the comments of all reviewers. They have improved the manuscript by moderating their claims to better align with the data and by incorporating additional experimental results. While I still find the allosteric modulation to be relatively mild, the revisions have strengthened the overall presentation. Given these improvements, I support publication in its current form.

Reviewer #3

(Remarks to the Author)

The authors have thoroughly answered to all my concerns and modified the manuscript accordingly. I thus recommend it for publication in Nature Communications.

REVIEWER COMMENTS

Reviewer comments are in *italics*. The responses are in blue.

Reviewer #1 (Remarks to the Author):

Turner et al. presents in this manuscript a structural study of Mtb 20S proteasome core particle by cryo-EM and HDX-MS. The major contribution of this work is that the authors were able to reconstruct a so-called “inactive” state of the Mtb 20S complex, with a few detailed structure features distinguishing this state from those previously reported. This work potentially provides new insights into the allosteric regulation of the Mtb 20S and paves the way for structure-based therapeutic discovery against Mtb. This work might be considered for publication if the authors can address the following issues.

We greatly appreciate the time this reviewer invested in our manuscript and their constructive criticism. We have worked diligently to address the issues raised.

(1) The naming of so-called “inactive” state of their newly observed 20S conformation may not appropriate. The authors appear to insufficiently understand the current stance and literature of the proteasome field and over-simplifying the process of proteasome activation into a two-state “switch”-like process. Almost all major literature regarding proteasome activation were missing and ignored in the references section. The 20S core particles have been resolved in “inactive” states that were inhibited by small molecules at the active sites, although structural insights into the catalytically active state were gleaned through these inactive inhibited 20S structures. More importantly, the 20S CP activation is now much better understood as a multi-step process, which involves pro-peptide cleavage at the active site, assembly of RP onto CP for priming but not immediately activation, and then substrate engagement, followed by deubiquitylation and large conformational transition of RP before the proteasome can be fully activated (the authors are referred to the recent literature of substrate-engaged human and yeast proteasome structures in Nature and Science). In eukaryotes, this process is further orchestrated by proteasome-associated proteins, like USP14/Ubp6 and UBE3C/Hul5. In many published works, even the 26S proteasome in a ground, resting state is considered “inactive”. Therefore, the claim that the “inactive” state of Mtb 20S CP in this work is the first inactive state of 20S ever discovered does not make much sense and represents a substantial ignorance and misunderstanding to the existing literature and knowledge accumulated in the proteasome field. From what the authors described, the so-called “inactive” state might be more like a “self-inhibited” or “auto-inhibited” state, where the switch helices block the access of substrate.

We appreciate the reviewer’s comments on the rich literature of the more complex eukaryotic proteasome system compared to its bacterial counterpart. In light of this, we have revised our manuscript to remove any assertions that the inactive state of *Mtb* 20S CP presented in our work is the first of its kind discovered. Instead, we now focus exclusively on the *M. tuberculosis* proteasome 20S and limit our claims to the switch helices of the *M. tuberculosis* system only. We also cite the works mentioned above as a preamble to our study so as to establish the current understanding of the 20S allostery as a whole:

“Both eukaryotic and *Mtb* proteasome systems are modulated by allostery^{12,14,20,21}. Eukaryotic 20S CP activation involves several steps²² including pro-peptide cleavage at the active sites^{23,24}, RP assembly²⁵, substrate engagement, deubiquitylation, and significant conformational changes in RPs, all essential for full activation^{26,27}. This process is further regulated by proteasome-associated proteins such as USP14/Ubp6 and UBE3C/Hul5^{28,29}.”

As per the reviewer’s suggestion we have renamed our 20S_{OFF} state to 20S_{auto-inhibited} state in the manuscript and in this response letter.

(2) *Following the above major concern, the authors should also avoid naming the state of 20S_{on} as “active” state. It may be better called “canonical resting state”.*

As per the reviewer’s suggestion and we have now renamed our 20S_{ON} state the “canonical resting state” abbreviated as 20S_{resting}, throughout the manuscript and in this response letter.

(3) *On line 421-423, the sentence reads confusing or inappropriate in that there is completely no data shown in this work that the so-call 20S_{off} conformation and associated mechanism is conserved across the three domains of life. Extrapolating structural observation on *Mtb* to a broader context, in this case, should be very careful and should adhere to the data and evidence. The heteroheptamers of the CP in eukaryotes are considerably more complex and well differentiated as compared to the homoheptamer assembly of *Mtb* 20S. This reviewer cannot believe that anything observed on *Mtb* 20S must be conserved in eukaryotes unless there is solid evidence.*

We have edited this sentence, and we no longer make any claims regarding the conservation of the mechanism described in our work. This revised text now reads:

“Our structural insights advance our understanding of the functional dynamics of *Mtb* 20S CP. Molecules that selectively bind to and stabilize the 20S_{auto-inhibited} of the proteasome established in this study, rather than the well-known canonical resting state (20S_{resting}) could be critical to future inhibitor development efforts that focus on allosteric rather than orthosteric sites of *Mtb* 20S CP. Aspects of the mechanism described here appear to be conserved in other structurally homologous systems. For example, allosteric activation of HslV protease, the *E. coli* homologue to the proteasome and the ancestor of archaeobacterial and eukaryotic 20S proteasomes⁵⁷, involves the rotation of switch helices to correctly position the active site (Fig. 7), similar to what Losick and colleagues have proposed for the protein phosphatase 2C superfamily⁴². Interestingly, the binding of RPs or substitutions at the α - β interface of the archaeal *T. acidophilum* 20S CP modulate substrate specificity and alter the cleavage products³⁷. These likely result from changes in the topology of the substrate binding pocket of *T. acidophilum* 20S CP due to the movement of the switch helices. Therefore, the mechanism described here may also apply

to other structurally homologous systems, though, this needs to be experimentally tested.”

(4) *The authors described that they used 3DVA to reveal the conformational variation. Can they show the 3DVA results in a statistical distribution of particles mapping to 2 or 3 dimensions of principal components in one of the Extended Data Figure? Can they show their classification of the 20 frames along one of the principal components they choose to visualize the conformational changes?*

As per the reviewer’s suggestion, we have added several figure panels (Supplementary Fig. 16) demonstrating the distribution of particles along selected principal components (1-3) of the 20S β _{T1A} dataset, and showing the subset sampling (without overlap) of the particles into 20 frames along principal component 3 that showcase the specific movement of the switch helices. The revised text and figure are:

“The third component of 3DVA represents the flexibility of the density corresponding to switch helices I and II and is unique to the 20S β _{T1A} variant (Supplementary Fig. 15 and Supplementary Movie 3). Accordingly, we sorted particles from the 20S β _{T1A} dataset into 20 intermediate reconstructions along the third principal component from 3DVA and refined them locally to produce maps with resolutions 2.5-2.7 Å (Supplementary Table 2). We subsequently fit our molecular models to the first and last intermediate reconstructed maps and achieved excellent refinement statistics (Supplementary Table 1). Frames 1 and 20 represent the most distinct states in the continuum (Supplementary Fig. 16). The comparison between the initial and final frames clearly illustrates the transition from a disordered state to a well-ordered conformation that closely mirrors with the canonical resting state of the switch helices.”

(5) *Although the authors did in vitro test of their 20S_{off} conformation model, there is no data to clarify if the 20S_{off} conformer is physiologically relevant or did exist in cells. Even indirect evidence should be helpful in this case.*

We appreciate the reviewer’s concern regarding the physiological relevance of the 20S_{auto-inhibited} state. It is indeed challenging to directly observe subtle conformational changes within cells due to limitations in state-of-the-art *in vivo* imaging technologies such as electron cryotomography. The *Mtb* cell thickness and the limited number of 20S CPs in intact cells (unlike for example ribosomal complexes) would limit our ability to reconstruct high-resolution maps under native cellular conditions. Our *in vitro* approach is necessary for observing these subtle conformational shifts, such as those in the switch helices and the collapse of the S1 pocket. We have revised the Discussion to reflect the point raised by this reviewer:

“While our findings are based on *in vitro* experiments, they provide a foundational understanding of potential allosteric mechanisms that could operate under physiological conditions. Our work brings the mechanistic understanding of the *Mtb* proteasome system closer to the well-established

archaeal and eukaryotic proteasomes and other structurally homologous systems.”

Furthermore, we used our structural data, complemented by additional site-directed mutagenesis, to probe the significance of the 20S_{auto-inhibited} state in the catalytic cycle of the enzyme that indirectly supports the potential biological relevance of the 20S_{auto-inhibited} state. Please refer to point 6 of Reviewer #2’s comments.

(6) Alternatively, the authors need to conduct validation experiments to prove that the 20S_{off} conformer is not caused by inappropriate biochemical treatment of proteasome samples and is not an artificially triggered, destabilized proteasome conformation that mostly does not exist along the functional pathway of the proteasome.

We have added intact protein mass spectra that show our 20S constructs are not chemically modified throughout the sample preparation step:

“We verified the molecular weight of the individual α - and β -chains using intact protein mass spectrometry (Supplementary Fig. 2).”

Supplementary Figure 2. Deconvoluted mass spectra of intact 20S CP variants reveal expected molecular weights for each subunit of interest (α_{WT} , α_{OG} , β_{WT} and β_{T1A}) within 1 Da. Spectra were summed over the two chromatographic peaks corresponding to the elution of each respective α - and β -subunit prior to deconvolution. Measured molecular weights and associated subunit are labeled.

In addition, we performed further 3DVA on our cryo-EM data for all of our 20S constructs.

This new analysis showed that the 20S_{WT} and 20S_{OG}, but notably not the ixazomib-bound 20S_{WT} variant undergo similar conformational changes as the T1A variant. Of note, this conformational change resembles the 20S_{auto-inhibited} state with a few notable differences. We have edited the text to reflect these points:

“Notably, when we applied 3DVA to both the apo 20S_{WT} (Supplementary Movie 4) and 20S_{OG} (Supplementary Movie 5) variants, we observed similar pattern of conformational changes as in the 20S_{βT1A} variant, with the C terminus of switch helix II unwinding (Supplementary Fig. 17). Here switch helix I exhibits notably reduced motion compared to the 20S_{βT1A} variant. The binding of ixazomib to 20S_{WT} stabilizes an on-pathway conformation that closely mirrors 20S_{resting}. As such, ixazomib-bound 20S_{WT} does not undergo the conformational change described above (Supplementary Movie 6). Overall, our 3DVA and HDX-MS data indicate that alternative conformation of the switch helices, though stabilized by the βT1A substitution, also naturally occurs in the wild-type 20S CP during its natural reaction cycle.”

Supplementary Figure 17. 3D variability analysis revealed density changes in Switch Helix II resembling those seen in (a) the $20S_{\beta T1A}$, (b) $20S_{WT}$, and (c) $20S_{OG}$ structures, but not in (d) the $20S_{Ixzazomib-bound}$ variant. Maps generated at negative (initial frame, coloured blue) and positive (final frame, coloured purple) latent coordinates along the variability components are overlaid. Switch Helix II of the β -subunit is highlighted with red arrows. See Supplementary Movies 3-6 for viewing the full motion.

(7) As mentioned above, allosteric regulation of the proteasome CP is a lot more complicated than just regulating the 20S core particle's proteolytic active site. This paper only addresses a very limited aspect of the allosteric regulation of the 20S CP in *Mtb*, and there is no evidence of 20S^{off} conformers will ever be observed in any species like yeast or human cells. Therefore, it is highly recommended that the authors revised the title of the paper to be specific and adhere to their core finding rather than generally claiming that they "finally" discover the structural basis for allosteric regulation of the proteasome CP "once for all", which we know it does not. Something like "Structural basis for auto-inhibition of active sites and its allosteric regulation in *Mtb* 20S proteasome core particle" would better serve the scholarly communication in this case.

We agree with the reviewer and have now edited our title to focus exclusively on the *Mtb* system. Our title now reads:

"Structural basis for allosteric modulation of *M. tuberculosis* proteasome core particle"

Reviewer #2 (Remarks to the Author):

In this manuscript, the authors investigate the structure of the Mycobacterium tuberculosis 20S proteasomal core particle in the absence of a regulatory particle using single-particle cryo-EM and hydrogen-deuterium (H/D) exchange experiments. Proteasomes in all domains of life are gated against random substrate access to the proteolytic chamber. Wild-type 20S particles degrade only small peptides that can diffuse into the proteolytic chamber without gate-opening, whereas larger peptides and proteins require regulatory complexes that associate with the 20S ends to facilitate gate opening and substrate entry.

Turner et al. determined four structures: wild-type 20S (20S^{wt}), a mutant where the active site threonine is substituted with alanine (20S T1A), an open-gate variant with the first seven residues of the alpha subunit deleted (20S OG), and 20S bound to a peptidyl boronate inhibitor (20S ixazomib). The authors propose that the structure of the mutant 20S proteasome, 20S T1A, represents an "inactive" or "20S OFF" state that prevents substrate binding based on three observations: the WT 20S and the 20S OG have mildly different affinities and Hill coefficients for peptide substrates, there is a small shift of a helix in the vicinity of the active site observed in the T1A mutant structure, mutations in the alpha subunit distant to the active site lead to a modest reduction in the rate of the reaction and the affinity for the substrate.

Although the experimental data is technically sound, the manuscript suffers from overinterpretation, and the conclusions are not justified by the experimental evidence. Unless the authors can provide convincing experimental evidence that the observed conformation is indeed relevant for the activity of the wild type enzyme, the manuscript is not a strong candidate for further consideration.

We thank the reviewer for raising these issues and for their thoughtful suggestions.

Below, we address these comments point-by-point, along with the additional feedback provided.

Below are some specific points of concern:

1. The abstract claims that "Both eukaryotic and Mtb proteasome systems are allosterically regulated, yet the specific conformations involved have not been captured in high-resolution structures to date. Here we present the first structure of ... any 20S CP in an inactive state ... distinguished from the canonical active state... by the conformation of switch helices I and II. The rearrangement of these helices collapses the S1 pocket, effectively inhibiting substrate binding."

These statements imply a regulatory mechanism that restricts substrate access, whereas such access is primarily controlled by the alpha-ring gating and interactions with regulatory particles. The 20S wt particle, in the absence of a regulatory complex, does not exhibit an "OFF" state. In fact, all previously solved structures of 20S wt, as well as the one in this study, are in the "ON" state. Furthermore, the purported "OFF" conformation does not prevent substrate binding; rather, it transitions to the "ON" state upon substrate binding (the authors describe a dynamic fluctuation between these two states). This structural plasticity is typical of enzymes during their reaction cycle, but does not constitute a regulatory "switch."

The reviewer makes an excellent point regarding the claim that the 20S allosteric changes we uncovered play a regulatory role. The primary objective of our manuscript was to establish a structural and conformational foundation regarding the role of allostery in the function of the 20S proteasome, as noted by the leading experts in the field cited in the Introduction. As noted above in response to Reviewer #1, we have significantly edited our manuscript and no longer make any claims regarding the regulatory nature of allosteric conformational changes. Instead, we focus on allosteric *modulation* of 20S activity. We have modified the **entire manuscript** to reflect this change. For brevity, we provide a few examples below. Our title now reads:

"Structural basis for allosteric modulation of *M. tuberculosis* proteasome core particle"

in the Results we write:

"...further highlighting the role of the switch helices in modulating the activity of *Mtb* 20S CP."

and in the Discussion we write:

"The fourth variant, however, provided the first example of *Mtb* 20S CP with the switch helices at the α - β interface in a conformation that completely obstructs the S1 substrate binding pocket, thereby modulating 20S activity."

2. Previous high-resolution structures of 20S CPs, including the crystal structure of the

20S T1A mutant (Li et al., 2010), have consistently shown the conformation that the authors describe as "ON." How do the authors explain the divergence between their 20S T1A structure and the one published by Li et al.?

The reviewer makes a pertinent observation. The divergence noted between our β -T1A structure and that described by Li et al. can likely be attributed to differences in proteasome maturation and experimental conditions. The structure by Li et al. includes the β -propeptide, whereas our T1A variant was purified without the β -propeptide. Additionally, X-ray crystallography, used by Li et al., stabilizes specific states of proteins in the crystalline environment, which may not capture alternate conformations accessible through cryo-EM. We have added the following text to the Discussion section to help explain these differences and address the reviewer's point:

"In contrast to the β T1A variant structure reported by Li et al., which was crystallized with the β -propeptide present and adopts the canonical state of the 20S, our cryo-EM structure of the 20S_{T1A} variant lacks the β -propeptide, potentially accounting for the observed conformational differences. The crystalline environment can influence the backbone conformation, side-chain flexibility, and the arrangement of water molecules and therefore may not fully represent the diversity of conformations the 20S can adopt in solution⁵⁶."

3. The authors claim that peptidyl boronates mimic the transition state of hydrolysis and were used in the hope of stabilizing "inactive" conformations. However, transition-state mimics should theoretically stabilize on-pathway conformations. Can the authors clarify this statement, particularly considering that their ixazomib-bound structure corresponds to the "ON" state, rather than an inactive state?

We agree that this sentence was poorly worded. We have edited it and now reads:

"Peptidyl boronates, such as ixazomib - a clinically approved human 20S CP inhibitor - represent a class of compounds that competitively inhibit both eukaryotic and prokaryotic 20S proteasomes by mimicking the transition state that the enzyme stabilizes during substrate hydrolysis. This characteristic makes ixazomib ideal for probing the allosteric mechanisms of the *Mtb* 20S CP, as it can stabilize on-pathway conformations that are otherwise transient and elusive in structural studies^{4,44,45}."

4. The authors do not observe an "OFF" state for 20S wt, nor do they identify changes in the alpha subunit conformations between the "ON" and "OFF" states. Yet, in lines 190-200, they argue that the decreased local resolution in the 20S wt alpha rings suggests interconverting conformations and, by extension, an allosteric network that spans to the alpha subunits. These conclusions are an overinterpretation and are not supported by the data. Furthermore, it is not a new concept that an allosteric network exists between the alpha and beta subunits, but the data provided here do not contribute new information about the nature of this allosteric network between alpha and beta subunits.

We agree that this sentence did not contribute positively to the manuscript, and we have

fully removed the previous claim about the interconverting conformations suggested by the local resolution in the α -rings.

5. *The H/D exchange experiments reveal changes in flexibility and local conformation between the 20S variants (wt, T1A, and ixazomib-bound), but they do not provide sufficient structural evidence for defining the allosteric network. Similarly, the cryo-EM structures show congruent alpha subunit conformations across all four variants. The experimental evidence does not support the conclusions.*

We appreciate the reviewer's thoughts and understand the concerns regarding our interpretation of HDX-MS and cryo-EM data. However, we respectfully disagree with this point. HDX-MS is a widely recognized method for assessing allosteric communication in high-molecular weight systems (see *Chem. Rev.* 2022, 122, 8, 7562-7623). We must emphasize that HDX-MS and cryo-EM complement each other by providing distinct insights into protein structure and dynamics. This complementarity is critical, as it broadens our understanding beyond static structures to include dynamic changes within the proteasome that extends beyond the previously vague notion of an allosteric network between the α - and β -subunits in *Mtb* 20S CP. Our HDX-MS data are of the highest quality and robustly characterize changes in backbone dynamics across our 20S variants. Importantly, HDX-MS excels in detecting subtle changes in flexibility and local conformations that are not always evident in cryo-EM data. For example, while cryo-EM did not show significant differences in the α - and β -subunits across our 20S $_{\beta T1A}$ and 20S $_{OG}$ variants, respectively, HDX-MS detected spatially resolved variations that clearly indicated allosteric interactions (Fig. 3).

We have revised our manuscript to clarify these points further, integrating more detailed descriptions of HDX-MS and moving additional data from the Supplementary Information into the main text to underscore the robustness and relevance of our findings. In the Results we write:

“HDX-MS is a powerful approach for assessing protein dynamics and structure by measuring the rate at which backbone amide hydrogens exchange with the deuterium from a D₂O-based buffer^{46,47}. The exchange rate depends on the local structure and dynamics of individual protein segments. Tightly hydrogen-bonded regions exchange slowly, while solvent-exposed and flexible regions exchange rapidly⁴⁸. In continuous-labeling HDX-MS, a sample of protein is diluted into a D₂O-based buffer, and deuteration kinetics are monitored as a function of D₂O exposure time. Backbone amides are categorized as either “closed” (H-bonded and exchange-resistant) or “open” (solvent-accessible and exchange-competent). Closed amides can exchange only through transient conformational fluctuations, while open sites undergo HDX rapidly⁴⁹. Ligand interactions often result in reduced deuterium uptake at residues involved in binding, while allosteric effects may manifest as increased or decreased uptake in distal regions⁵⁰. This approach allows detailed mapping of protein-ligand interactions and conformational changes across the protein structure.”

Later in the Results section we write:

“Taken together, these HDX-MS results highlight regions that facilitate communication across the α - β interface into the switch helices I and II that in turn modulate the activity of the 20S CP catalytic triad on the β -subunit ~ 50 Å away from the gating residues. These data also highlights the synergy and complementarity of cryo-EM and HDX-MS in probing supra-molecular systems⁵¹. The structure of the catalytic β -subunits of *Mtb* 20S CP in the 20S_{OG} state compared to 20S CP_{WT} are remarkably consistent across published work^{4,11} and in our study (Fig. 2a). And yet, the HDX data unexpectedly shows significant changes in α R26 in 20S _{β T1A} variant and multiple regions of the β -subunit in the 20S_{OG} variant, far beyond local effects near substitution sites (Fig. 3). These data compellingly demonstrate that HDX-MS is capturing allosteric communication across the 20S CP, illustrating changes that are not merely local but indicative of broader structural interactions.”

Figure 3. Heat map of relative deuterium uptake highlights the allosteric pathway between the α - or β -subunits of *Mtb* 20S CP. Changes in deuterium uptake of each state relative to the 20S_{WT} variant are depicted for each peptide generated in both the (a) α -subunit and the (b) β -subunit. Only changes larger than 0.5 Da were considered significant and were coloured according to the colour bar (bottom). The peptides generated for each subunit

are listed across the top of the heat map and D₂O exposure times are listed along the side. Gray squares indicate absent data. Peptides associated with the RP-binding sites of the α -subunit and switch helices of the β -subunit are labeled.

The utility of HDX-MS and its complementarity with cryo-EM is further illustrated by our new 20S variants that we added in the revisions. We used our cryo-EM structures to design substitutions, used HDX-MS to validate its impact on the conformational dynamics of the 20S without necessitating additional structural determination, and used biochemical assays to read out its activity. This is highlighted in the Results section:

“We used pulsed HDX-MS to compare the structure of several 20S variants to 20S_{WT} in the apo state. The deuteration pattern of the 20S _{α Q98K} variant in the apo state was indistinguishable from that of 20S_{WT}, except for the C terminus of switch helix II which was more deuterated compared to 20S_{WT} (Supplementary Fig. 19a). This is consistent with the α Q98K substitution affecting the conformation of switch helix II via long-range allosteric communication. The β A92-94G substitution also destabilized switch helix I via the loss of local interactions. The β A100S substitution had increased deuterium uptake in several regions in the α - and β -subunits, including in switch helices I and II, in peptides encompassing the catalytic residues, and in residues in the α -subunits involved in RP binding and the α - β interface. These data affirm that the substitutions exert their intended effects on the structure and dynamics of *Mtb* 20S CP. Next, we tested the ability of these variants in binding ixazomib. Similar to 20S_{WT}, all variants showed protection in catalytic residues and in switch helix I upon ixazomib binding. The 20S _{α Q98K} variant, which was active in our peptidase assays, also showed protection switch helix II in response to ixazomib binding, indicating that the impact of the substitution can be reversed via the addition of small molecules that favour the canonical resting state of 20S CP. However, the 20S _{β A92-94G} variant, which was inactive in our assays, did not undergo protection upon ixazomib binding in the C-terminal portion of switch helix II, further highlighting the role of the switch helices in modulating the activity of *Mtb* 20S CP.”

Supplementary Figure 19. The 20S variants designed to stabilize the 20S_{auto-inhibited} state retain the capacity to bind substrate. Pulsed HDX-MS experiments were used to explore the conformational dynamics of the less active 20S_{αQ98K}, and inactive 20S_{βA93–94G} and 20S_{βA100S} variants in the apo and ixazomib-bound states. Heat maps of relative deuterium uptake against 20S_{WT} were generated, where only changes larger than 0.5 Da were considered significant and were coloured according to the colour bar (bottom). **(a)** Each variant shows increased conformational dynamics in one or both Switch Helices compared to the 20S_{WT} in the apo state. **(b)** The variants show similar D-uptake patterns to 20S_{WT} in the ixazomib-bound state, except for a single peptide in Switch Helix II of the 20S_{βA100S} variant, signifying that they can still bind substrate. The peptides generated for each subunit are listed across the top of the heat map and the associated variants are listed along the side. Gray squares indicate absent data. Peptides associated with the RP-binding sites of the α-subunit and switch helices of the β-subunit are labeled.

6. Point mutations in the alpha subunits and beta subunits were generated to support the existence of an inactive state. The authors argue that the v_{max} of peptide cleavage at the beta subunits can be influenced by mutations in the alpha subunits. However, the observed changes are very mild (v_{max} changes from 3 to 4 μM , with K_{05} changes from 95 to 60 μM) and do not support a regulatory “activity switch”. The mutations introduced in the beta subunits (Y35F and V53Q) likely directly disturb active site residues, since Y35

H-bonds to Lys33 in the active site and to Val53. It is therefore not surprising to observe changes in catalytic parameters.

We appreciate the reviewer's concern regarding the implications of the observed catalytic changes due to mutations in the α - and β -subunits. In the revised manuscript, we used our cryo-EM structures to **design several new variants** that further illustrate the role of the switch helices in the activity and allostery of the 20S CP. One of these variants in the α -subunit is 37 Å away from the active sites and reduces the activity three-fold. We have significantly revised this section with new data and text:

“Mutations actuate an allosteric population shift in *Mtb* 20S CP

To investigate the role of individual residues in the allosteric network of *Mtb* 20S CP, we calculated pairwise deviations between the 20S_{auto-inhibited} and 20S_{resting} structures and identified regions with significant conformational differences (Supplementary Fig. 18a, b). We targeted these regions using single-residue substitutions in either the α - or β -subunit. While the deviations in the α -subunit based on our atomic models are small, our HDX-MS data clearly show allosteric communication in structural elements at the α - β interface and around the RP binding site (Fig. 3 and 4). We identified a key α -subunit loop at the α - β interface that undergoes substantial rearrangement between the 20S_{auto-inhibited} and 20S_{resting} states. Specifically, α D93 rotates by ~80 degrees between these two states (Supplementary Fig. 18c). We noted that the α Q98K substitution would selectively stabilize the 20S_{auto-inhibited} conformation of α D93, providing a potential target for modulating 20S CP activity via an α -subunit site 37 Å away from the catalytic sites. In the α -subunit, we also targeted the RP-binding pocket with two variants, 20S _{α S17F} and 20S _{α K52F}, chosen based on a previous study³⁴ where similar substitutions in the archaeal 20S CP mimicked HbYX binding and induced gate-opening (Fig. 6). We hypothesized that these mutations facilitate communication across the α - and β -subunits, which is crucial for proteasome activation.

Structural deviations between the 20S_{auto-inhibited} and 20S_{resting} structures were more pronounced in the β -subunit and occurred particularly in switch helices I and II, and in adjacent structural elements (Supplementary Fig. 18b). For the β -subunit, we targeted β Y35 positioned between switch helix I and the active site. In the 20S _{β Y35F} variant, the mutation from Y to F results in the loss of a hydroxyl group essential for H-bonding with the catalytic β K33 and β V53 on switch helix I (Fig. 6). We expected that this substitution would likely weaken the interactions critical for maintaining the active conformation of the switch helices and the S1 pocket of the active site, thereby reducing the activity of the 20S CP. Conversely, in the 20S _{β V53Q} variant would reinstate a β - β intra-ring H-bond found in the archaeal system but absent in *Mtb* 20S CP, reasoning that this would enhance activity by stabilizing the active conformation of the switch helices. Additionally, we explored the C terminus of switch helix II, which shows unwinding in 20S_{auto-inhibited}. We hypothesized that the α Q98K substitution via long-range allosteric communication, and the β A92G- β A93G- β A94G substitution (hereafter referred to as β A92-94G) which swaps helix-stabilizing alanine residues for glycines, would both destabilize switch helix II and favour

the 20S_{auto-inhibited} conformation and reduce activity. Lastly, β A100 is positioned at the N terminus of a β -strand between switch helices I and II in the 20S_{resting} structure, and has elevated RMSD between the two states (Supplementary Fig. 18b). The preceding residues are disordered in the 20S_{auto-inhibited} state. We sought to stabilize the disordered conformation of switch helix II with a β A100S substitution (Fig. 6). This substitution provides an additional hydrogen bond with the backbone of β L101 in the 20S_{auto-inhibited} conformation and was also expected to reduce activity.

We used size exclusion chromatography to purify samples of correctly assembled oligomeric particles of all 20S variants. Steady-state kinetic analysis of Z-VLR-AMC hydrolysis by the variants yielded sigmoidal curves, indicative of positive cooperativity. Fitting of the data to the Hill model yielded Hill coefficients ranging from 1.7 to 2.2 (Fig. 6, Table 1, and Supplementary Fig. 3). The 20S _{α Q98K} variant that we designed to favour the auto-inhibited conformation, had a 3-fold reduced V_{max} value of 1.0 $\mu\text{M s}^{-1}$ (95% C.I. 0.9 – 1.2) relative to 20S_{WT}, with statistically insignificant changes in $K_{0.5}$ values (95% C.I. 59.0 – 88.5). The α -subunit variants, 20S _{α S17F} and 20S _{α K52F}, showed $K_{0.5}$ values of 57.5 μM (95% C.I. 43.6 – 75.9) and 59.3 μM (95% C.I. 50.5 – 69.8), respectively, while the β -subunit variants, 20S _{β Y35F} and 20S _{β V53F}, displayed increased $K_{0.5}$ values of 68.5 μM (95% C.I. 58.5 – 80.2) and 86.9 μM (95% C.I. 65.8 – 117.0), respectively. Interestingly, while the V_{max} of 20S _{α S17F} matched those of the 20S_{WT} and 20S_{OG} variants at 3.2 $\mu\text{M s}^{-1}$ (95% C.I. 26 – 3.9), the 20S _{α K52F} and 20S _{β V53Q} variants exhibited increased V_{max} values of 4.2 $\mu\text{M s}^{-1}$ (95% C.I. 3.7 – 4.7) and 5.9 $\mu\text{M s}^{-1}$ (95% C.I. 4.9 – 7.2), respectively. In contrast, the 20S _{β Y35F} variant had a reduced V_{max} of 1.4 $\mu\text{M s}^{-1}$ (95% C.I. 1.3 – 1.6). The switch helix II variants 20S _{β A92-94G} and 20S _{β A100S} that we designed to favour the auto-inhibited conformation were inactive. We next combined the most active variants of the α - and β -subunits to determine if the effects of the activating mutations were additive. The 20S _{α K52F/ β V53Q} variant maintained unchanged n and $K_{0.5}$ values compared to the other variants, yet it achieved the highest V_{max} , reaching 8.0 $\mu\text{M s}^{-1}$ (95% C.I. 6.2 – 10.4).

We used pulsed HDX-MS to compare the structure of several 20S variants to 20S_{WT} in the apo state. The deuteration pattern of the 20S _{α Q98K} variant in the apo state was indistinguishable from that of 20S_{WT}, except for the C terminus of switch helix II which was more deuterated compared to 20S_{WT} (Supplementary Fig. 19a). This is consistent with the α Q98K substitution affecting the conformation of switch helix II via long-range allosteric communication. The β A92-94G substitution also destabilized switch helix I via the loss of local interactions. The β A100S substitution had increased deuterium uptake in several regions in the α - and β -subunits, including in switch helices I and II, in peptides encompassing the catalytic residues, and in residues in the α -subunits involved in RP binding and the α - β interface. These data affirm that the substitutions exert their intended effects on the structure and dynamics of *Mtb* 20S CP. Next, we tested the ability of these variants in binding ixazomib. Similar to 20S_{WT}, all variants showed protection in catalytic residues and in switch helix I upon ixazomib binding. The 20S _{α Q98K} variant, which was active in our peptidase assays, also showed protection switch helix

II in response to ixazomib binding, indicating that the impact of the substitution can be reversed via the addition of small molecules that favour the canonical resting state of 20S CP. However, the 20S $_{\beta A92-94G}$ variant, which was inactive in our assays, did not undergo protection upon ixazomib binding in the C-terminal portion of switch helix II, further highlighting the role of the switch helices in modulating the activity of *Mtb* 20S CP.”

Figure 6. The allosteric equilibrium of *Mtb* 20S CP can be modulated by targeting residues in either the α - or β -subunit. Leveraging insights from both the cryo-EM and HDX-MS studies, a series of point mutations were made to target areas identified as being allosteric hotspots. (a) Single-residue substitutions to either the α - or β -subunits resulted in modulated catalytic function; (b) Residues S17 and K53 of the α -subunits were targeted for mutation due to their role in RP-binding and proteasome activation. In a

previous study using the archaeal 20S CP, the substitution of these residues with a F residue induced gate-opening; and (c) α Q98 was targeted as a likely allosteric modulator across the α/β -interface. β Y35 and β V53 were targeted due to their proximity to the active site and switch helices. The Y35F substitution was made to prevent sidechain-mediated H-bonds that stabilize the active site. V53Q mutation was made to restore a H-bond across intra-ring β -subunits as seen in the archaeal system. (d) A92 – 94 and A100 of the β -subunits were targeted to stabilize the loop conformation of Switch Helix II.

Supplementary Figure 18. Alignment of the 20S_{resting} and 20S_{auto-inhibited} states reveals the residues that undergo the greatest structural shifts between the two conformations. α RMSD scores showing the local structural differences between 20S_{resting} and 20S_{auto-inhibited} states in the (a) α -subunit or (b) β -subunit. The top 5% of residues showing the largest RMSD values are labeled with the respective residue number. A dashed line indicates the cutoff value separating the top 5% from the remaining datapoints; and (c) Between the 20S_{resting} and 20S_{auto-inhibited} states, α D93 undergoes a $\sim 80^\circ$ rotation. This residue is a likely point of allosteric communication across the α/β -interface.

Supplementary Figure 19. The 20S variants designed to stabilize the 20S_{auto-inhibited} state retain the capacity to bind substrate. Pulsed HDX-MS experiments were used to explore the conformational dynamics of the less active 20S _{α Q98K}, and inactive 20S _{β A93–94G} and 20S _{β A100S} variants in the apo and ixazomib-bound states. Heat maps of relative deuterium uptake compared against 20S_{WT} were generated, where only changes larger than 0.5 Da were considered significant and were coloured according to the colour bar (bottom). **(a)** Each variant shows increased conformational dynamics in one or both Switch Helices compared to the 20S_{WT} in the apo state. **(b)** The variants show similar D-uptake patterns to 20S_{WT} in the ixazomib-bound state, except for a single peptide in Switch Helix II of the 20S _{β A100S} variant, signifying that they can still bind substrate. The peptides generated for each subunit are listed across the top of the heat map and the associated variants are listed along the side. Gray squares indicate absent data. Peptides associated with the RP-binding sites of the α -subunit and switch helices of the β -subunit are labeled.

Variant	$K_{0.5}$ (μM)	95% C.I. $K_{0.5}$ (μM)	n	95% C.I. n	V_{max} ($\mu\text{M s}^{-1}$)	95% C.I. V_{max} ($\mu\text{M s}^{-1}$)
20S _{WT}	94.5	79.9 – 111.7	1.6	1.4 – 1.7	3.1	2.8 – 3.5
20S _{Og}	39.1	35.0 – 43.7	2.3	1.9 – 2.7	3.4	3.1 – 3.7
20S _{αK52F/βV53Q}	95.3	68.9 – 131.8	1.8	1.4 – 2.1	8.0	6.2 – 10.4
20S _{βV53Q}	87.8	65.8 – 117.0	1.7	1.3 – 2.0	5.9	4.9 – 7.2
20S _{αK52F}	59.3	50.5 – 69.8	1.9	1.6 – 2.2	4.2	3.7 – 4.7
20S _{αS17F}	57.5	43.6 – 75.9	1.8	1.3 – 2.2	3.2	2.6 – 3.9
20S _{βY35F}	68.5	58.5 – 80.2	1.9	1.6 – 2.2	1.4	1.3 – 1.6
20S _{αQ98K}	73.3	59.0 – 88.5	2.2	1.5 – 2.9	1.0	0.9 – 1.2
20S _{βA100S}	inactive					
20S _{βA92-94G}	inactive					

Table 1. Kinetic parameters and associated 95% confidence intervals of each 20S variant against Z-VLR-AMC.

7. The main established role of the 20S proteasome in *M. tuberculosis* is the degradation of pupylated substrates by the Mpa/20S complex. The relevant question is therefore, how the postulated “activity switch” affects degradation of bona fide protein substrates in the context of the full degradation complex. The authors do not provide any data on this. The authors should test, how their point mutants behave in degradation of pupylated substrates by the Mpa/20S complex.

The reviewer raises a valid point regarding the degradation of pupylated substrates by the Mpa/20S complex, which is central to the biological role of the *M. tuberculosis* proteasome. While exploring this with our variants may be insightful, executing this faces several challenges. *Mtb* 20S_{WT} is not active together with WT Mpa and requires the use of the open-gate variant of 20S for functional and structural studies in published work:

M. Kavalchuk, A. Jomaa, A. U. Müller, E. Weber-Ban, Structural basis of prokaryotic ubiquitin-like protein engagement and translocation by the mycobacterial Mpa-proteasome complex. Nat. Commun. 13, 276 (2022);

and

X. Xiansha, ... H. K. Darwin, H. Li, The β -Grasp Domain of Proteasomal ATPase Mpa

Makes Critical Contacts with the *Mycobacterium tuberculosis* 20S Core Particle to Facilitate Degradation. mSphere 0, e00274-22 (2022).

This complication makes it challenging to directly attribute observed effects to the mutations, as they could be influenced by the allosteric impact of gate removal and by the affinity of the variants for Mpa. Nonetheless, we took the reviewer's comment to heart, and performed the following experiments and added them to the Results:

“One of the primary roles of the proteasome system in mycobacteria is to degrade pupylated substrates^{10,53,54}. Mycobacterial proteasome activator (Mpa) recruits, progressively unfolds, and threads pupylated substrates into the 20S CP for degradation. We sought to assess the ability of our most active 20S variants in degrading a Pup-dihydrofolate reductase (DHFR) fusion substrate as part of the Mpa:20S CP degradation complex. *Mtb* 20S_{WT} with wild-type full-length Mpa failed to degrade Pup-DHFR (Supplementary Fig. 20); only the 20S_{OG} variant degraded Pup-DHFR. These observations are consistent with the literature, which describe the β -grasp domain of Mpa preventing effective binding with 20S CP_{WT}⁵⁵. Lastly, we tested the ability of the 20S _{β V53Q} variant in degrading Pup-DHFR. Despite its higher peptidase activity, we opted to exclude the α K52F substitution to avoid potential interference with Mpa binding. Even our second-most active variant (20S _{β V53Q}) showed no detectable activity against Pup-DHFR (Supplementary Fig. 20). While these activating substitutions could be introduced in the background of 20S_{OG}, the effects attributable to these substitutions may be obscured due to the dominant functional characteristics of the 20S_{OG} variant. This complexity would hinder the clear attribution of any observed changes in activity effects directly to the substitutions, as they could be potentially influenced by the affinity of the 20S variant-Mpa interaction.”

Supplementary Figure 20. The 20S_{OG}:Mpa complex is capable of degrading a pupylated, model substrate. SDS-PAGE analysis monitoring the degradation of the model substrate, Pup-EcDHFR, at 37 °C over 25 h reveals that (a) substrate degradation is Mpa-dependent; and (b, c) only the 20S_{OG}:Mpa complex is capable of degrading substrate. Timepoints were collected at the indicated hours and reactions were stopped through the addition of Laemmli buffer. Samples were separated on a 20% Tris-glycine gel with Blue Elf ladder (Froggabo) as reference.

We believe that our approach focuses on using a short peptidic substrate that directly probes the active site without depending on Mpa or any other regulatory particle. We believe this provide clearer, though more limited, insights into the 20S allosteric modulation. We acknowledge this limitation and have revised the Discussion section of

the manuscript:

“Our attempts to explore the impact of amino acid substitutions on Mpa-mediated degradation of pupylated substrates were complicated by the weak interaction between Mpa and the full-length 20S CP. As in previous work^{19,41}, stable interaction and effective degradation of pupylated substrates by *Mtb* Mpa-20S were only achieved *in vitro* with the 20S_{OG} variant lacking the first seven residues of the α -subunits. The introduction of mutations into the 20S_{OG} variant would mask the intrinsic effects of these mutations due to its dominant functional characteristics. This could complicate the attribution of observed effects solely to the substitutions, as function might also be influenced by the affinity of the variant-Mpa interactions. To avoid these complexities, our experiments utilize a short peptidic substrate that directly probes the proteasome's active site independent of RP interactions. While this approach limits our ability to fully capture the biological role of the *Mtb* proteasome, it offers more definitive evidence of allosteric modulation within the 20S CP.”

8. Lastly, the proposal of an “activity switch” implies that the 20S is in an inactive state and needs to be activated by a specific trigger. What is the mode of activation? Literature would suggest that it is the binding of regulatory particles. In that case, the wt 20S in isolation should remain inactive and should only be activated in presence of the regulator. This is however not what the authors observe, since their 20S wt exhibits an “ON” state and shows degradation activity towards the tripeptide substrate.

This is a valid point that to some extent is related to comment #1 from the same reviewer. Our earlier notion of an “activity switch” has been refined in our revised manuscript to better describe it as allosteric *modulation*, rather than implying a binary inactive to active state transition that necessitates a specific trigger. In this new context, the “mode of activation” as described in our original submission, is substrate or inhibitor binding to the 20S CP. The WT enzyme is not fully in the active form. The binding of substrates or active-site inhibitors such as Ixazomib shifts the equilibrium further towards the active 20S_{canonical} state from the 20S_{auto-inhibited} state. This directly follows from the Hill model that fits our data well.

Reviewer #3 (Remarks to the Author):

In this article, Turner and coworkers decipher the structural bases responsible for the allosteric regulation of the Mtb 20S core proteasome:

- They solved four near-atomic resolution (2.5 – 2.7 Å) structures of *Mtb* 20S particles by cryo-EM: apo and ixazomib bound WT 20S, 20S β T1A and 20S_{OG} variants.
- While the structure of WT bound to ixazomib shows a binding mode that is similar to that observed in human 20S, comparison of apo WT vs. 20S β T1A shows that the β T1A mutation is enough to slightly rotate the Switch Helix 1, which occludes the S1 pocket, impeding further binding of ixazomib. Importantly, this represents the first structure of an inactive form of the 20S proteasome.
- Substrate degradation comparison of WT vs β T1A vs OG conformers shows an

increased cooperativity of OG vs. WT.

- The comparison by HDX-MS of the apoWT with the ixazomib-bound-WT and OG & T1A mutants highlighted some differences in dynamics that could not be observed by cryo-EM.

- The discovery of mixed EX1-EX2 regimes (in the ixazomib bound WT and T1A mutant), characteristic of slow interconversion changes is nicely confirmed and illustrated by the 3DVA approach. Such analysis of the T1A mutant shed light on the interconversion of the switch helices I and II between two conformers that were assigned as active (on) and inactive (off) states of the 20S, respectively.

- Site-directed mutagenesis rationalized based on HDX-MS, cryoEM results and previous results from the literature allowed to further describe the allosteric dialogue between the alpha and beta rings. A double mutant showed that the positive effects observed on the alpha and beta rings could be additive. Finally, the fact that the longer substrate could not be proteolyzed by any of these mutants, nicely shows that these reciprocal allosteric changes are uncoupled with the opening of the gate.

Overall, this very well written and illustrated article confirms and further describes a general mechanism of proteasome bidirectional allostery observed throughout the evolutionary tree. Besides, the cryo-EM structures provided here, together with the HDX-MS analysis, provide striking evidence for a major role played by the switch helix 1 (beta subunit) in this allostery, conveying structural changes from the beta to the alpha ring, and reversely. The experiments were perfectly well conducted and the main results confirmed by multiple complementary approaches.

For all these reasons, this article deserves to be published in Nat Commun, with minor corrections, as detailed below:

We appreciate the reviewer's enthusiasm for our work and their constructive criticism that we address point-by-point below.

- In the introduction, the authors mention an article from S. Gygi and coworker, suggesting the presence of long-range allosteric regulation in Mtb 20S proteasome. However, no mention is made in the introduction to similar results/observations made in 20S proteasome for other prokaryotes (Ruschak et al, PNAS 2012), yeast & mouse (Arciniega, 2014, PNAS) or human (Osmulski and Gaczynska 2002, Cell / Rego and da Fonseca Mol Cell 2019 / Lesne et al, 2020 Nat Commun). These articles, showing allosteric regulation following binding of regulatory particles or binding/modification of the catalytic residues, include results coming from AFM, NMR, CryoEM and HDX-MS experiments. Most of these articles are actually discussed in the discussion, but reference should also (or alternatively) be made in the introduction, to present the state of the art in terms of 20S allostery, as a preamble to this study.

The reviewer raises a good point which mirrors comment #1 from Reviewer #1. We now discuss the rich literature of proteasome allostery in the introduction. In the Introduction we write:

“Both eukaryotic and Mtb proteasome systems are modulated by

allostery^{12,14,20,21}. Eukaryotic 20S CP activation involves several steps²² including pro-peptide cleavage at the active sites^{23,24}, RP assembly²⁵, substrate engagement, deubiquitylation, and significant conformational changes in RPs, all essential for full activation^{26,27}. This process is further regulated by proteasome-associated proteins such as USP14/Ubp6 and UBE3C/Hul5^{28,29}.”

- P4. L91: *did the authors try to assemble double mutant 20S β T1A/OG?*

As per the reviewer’s suggestion, we did attempt to assemble 20S β T1A/OG, and the complex indeed does assemble, albeit with lower efficiency compared to our other variants. Given that the variant lacks a catalytic β T1, we were not able to test for activity. We have added a few sentences to the Methods section to mention the changes in the 20S assembly yield in our variants.

“The 20S CP assembly yield was the highest for 20S_{WT}, followed by 20S _{β T1A}, 20S _{$\alpha\Delta$ 2-6}. The 20S _{$\alpha\Delta$ 2-6, β T1A} construct also assembled into mature particles, albeit with substantially lower yield.”

- P5. L114: *“The shift in the Hill coefficient and K0.5 values between the 20SWT and 20SOG variants points to increased cooperative interactions when the gating residues are removed”. The reviewer is wondering if this increase in the Hill coefficient is really due to a better affinity of the substrate for the catalytic sites or to a better accessibility to these sites (higher diffusion) thanks to the removal of the gate? Is it really the allosteric cooperativity that is increased or rather the steric hindrance that is decreased?*

This is an insightful thought from the reviewer. Our interpretation of increased allosteric cooperativity, rather than merely reduced steric hindrance due to gate removal, is supported by a reaction progress curves in a new figure (Supplementary Fig. 4). We used HydroPro to calculate diffusion coefficient of Z-VLR-AMC at room temperature. We then used this value to perform numerical simulations of particle diffusion, estimated using the Einstein-Smoluchowski equation, to further support our claim that allostery is likely the main cause of the changes in the kinetic parameters. We do, however, acknowledge that limited diffusion might be a contributing factor.

“The reaction progress curves for both the 20S_{WT} and 20S_{OG} variants display a rapid initial rate and are sensitive to changes in initial substrate concentration (Supplementary Fig. 4a, b)⁴³. These observations and the microscopic length scales involved (Supplementary Fig. 4c) suggest that the changes in kinetic parameters arise from allosteric modulation of the 20S proteasome due to gate removal, rather than from decreased steric hindrance caused by the absence of gating residues. Nonetheless, we concede that the role of gating residues as a diffusion barrier may also influence these changes.”

Supplementary Figure 4. Representative progression curves reveal that the gating residues do not inhibit Z-VLR-AMC degradation. (a) Reaction progression initially increases at a rapid rate and then decelerates for both $20S_{WT}$ and $20S_{OG}$ variants; (b) A 30-sec fitting window, indicated by the pink box in panel (a), was used to determine the initial velocities of substrate degradation. The calculated fits used to determine the rate of product formation are shown as black lines.; and (c) The root mean square distance $\sqrt{\langle x^2 \rangle}$ traveled by diffusing particles undergoing Brownian motion in time t , estimated using:

$$\sqrt{\langle x^2 \rangle} = \sqrt{2Dt}$$

which can be readily derived from the Einstein-Smoluchowski relation. We used a diffusion coefficient D of $3.2 \times 10^{-10} \text{ m}^2\text{s}^{-1}$, estimated for Z-VLR-AMC

using HydroPro¹ at room temperature, being reduced by 10-, 100-, and 1000-fold by the gating residues; and (d) The concentration profiles above a plane where a solute is diffusing. The curves represent the equation:

$$c(x, t) = \frac{n_0}{A(\pi Dt)^{1/2}} e^{-x^2/4Dt}$$

where each curve is labeled with different values of Dt . The units of Dt and x are arbitrary but chosen such that Dt/x^2 remains dimensionless. For instance, if x is measured in meters, Dt would be in square meters. On the microscopic length scale, for an ordinary diffusion coefficient $D = 10^9 \text{ nm}^2 \text{ s}^{-1}$, a Dt value of 1 nm^2 corresponds to $t = 1 \times 10^{-9} \text{ s}$. These plots demonstrate the effectiveness of diffusion over microscopic length scales. See ref².

- P5.L114: *“Together, these results support the view that the 20S CP is an allosteric system which interconverts between multiple conformations that differ in their activity and affinity for the substrate.” At this stage (WT vs. OG), the referee is not sure that this assertion can be made, since the increase in activity could only reflect an increase in accessibility for the substrate, which is not necessarily due to allosteric changes. The subsequent T1A mutation is probably a better example of long-range allosteric change. This sentence could there be placed later on in the manuscript.*

As per the reviewer’s recommendation, we have moved this sentence to the end of the section.

- P5.L125: *Did the authors try to measure the affinity and Hill coefficient of ixazomib for the OG variant?*

As per the reviewer’s recommendation, we performed additional experiments and now report the IC₅₀ and the Hill coefficient of ixazomib against the OG variant. The new text and Figure read:

“We found that ixazomib inhibited the peptidase activity of 20S_{WT} against Z-VLR-AMC. Fitting the dose-response curve to a modified Hill model (see Supplementary Information) yielded an IC₅₀ value of 1.1 μM (95% C.I. 0.9 – 1.2) and, notably, a Hill coefficient of 2.1 (95% C.I. 1.6 – 2.6) (Supplementary Fig. 6a, b, c).”

Supplementary Figure 6. The peptidyl boronate, ixazomib, functions as a competitive inhibitor against the 20S CP. (a) Dose-response curve depicting the inhibitory effect of increasing ixazomib concentrations against the hydrolysis of the small tripeptide substrate, Z-VLR-AMC, by $20S_{WT}$ fitted to the Hill model; (b) Histogram representing 100,000 Monte Carlo simulations used to determine the Hill coefficient and respective confidence intervals; (c) Histogram representing 100,000 Monte Carlo simulations used to determine the IC_{50} value and respective confidence intervals; and (d) Cryo-EM structure of the $20S_{WT}$ CP bound to ixazomib (teal) with density of the small molecule

represented as mesh. Density associated with ixazomib was noted in the active site of the CP and displays similar hydrogen bonding patterns to the previously solved structure of the human 20S CP bound to a related peptidyl boronate³. (e) Dose-response curve depicting the inhibitory effect of increasing ixazomib concentrations against the hydrolysis of the small tripeptide substrate, Z-VLR-AMC, by 20S_{OG} fitted to the Hill model; (f) Histogram representing 100,000 Monte Carlo simulations used to determine the Hill coefficient and respective confidence intervals; and (g) Histogram representing 100,000 Monte Carlo simulations used to determine the IC_{50} value and the respective confidence intervals.

- P8.L213: *The Nter of the β subunit is marked as covered in the peptide mapping (Extended Data Fig.6), even in the ixazomib bound WT. Since the binding is supposed to be covalent, the addition of the ligand will increase the MW of these Nter peptides: did you monitor the HX on these covalently modified peptides or on the portion of un-liganded Nter peptides? Similarly, the T1A mutation will also modify the MW of these Nter peptides, meaning that this region cannot be differentially analyzed. Could you comment on this?*

This reviewer's observation highlights an important consideration in HDX-MS analysis. Boron is a Lewis acid that forms non-ionic reversible bonds with nucleophilic groups. In our studies of proteases such as the proteasome and ClpP proteases, the binding of peptidyl boronates, such as ixazomib, is readily reversed upon enzyme denaturation, allowing the dissociation of the inhibitor from the enzymes and allowing us to monitor the deuterium uptake of unmodified peptides. For our 20S variants, including T1A, we specifically mapped and analyzed the deuterium uptake of peptides incorporating the mutated residue. We have edited the Methods section to clarify this:

"We performed additional peptide mapping experiments for 20S CP samples that included mutations as needed. This was not necessary for ixazomib-bound samples as ixazomib forms a reversible bond with the β -subunit and rapidly dissociates upon denaturation and leaves a chemically unmodified enzyme."

- P9.L218-227: *While the protection effect induced by ixazomib on the beta subunit is very clear, the two following assertions concerning subunit alpha seem a bit weaker and should be tuned down. "Of note, ixazomib binding also induced changes in the conformational dynamics of the α -subunit." These are very slight changes compared to the 1TA or OG mutants. Then, concerning residues alpha1-12: "These measurements are supported by multiple partially overlapping peptides". In Extended Data 6, only the 1st timepoint of peptide 1-12 is shown as deprotected, but none of peptides 1-6, 1-14, 7-12, so what are these overlapping peptides? This is clear for the protection of the Swith Helix I, but not for the slight changes observed in subunit alpha.*

As per the reviewer's suggestion, we have moderated the assertions regarding the changes in α -subunit dynamics upon ixazomib binding. The statement concerning the multiple partially overlapping peptides was indeed misplaced and has been removed.

- P9.L236-237: *“Likewise, peptides at the α - β interface also demonstrated reduced deuterium uptake, consistent with the changes induced by ixazomib binding. However, the changes span an entire helix in the α -subunit (peptides containing α 75-90) rather than just those residues localized to the site closest to the β -ring.” This sentence seems to suggest that the interface is more protected in the 1TA mutant vs. in the ixazomib sample, which is actually the contrary, since in both Extended Fig.6 and 7, the peptides from subunit beta protected in 1TA at the alpha/beta interface are less numerous (87-98) than in the ixazomib sample (85-91/87-91/87-98/99-104/99-110).*

We agree with the reviewer that this statement is confusing. The edited text now reads:

“Likewise, peptides at the α - β interface showed reduced deuterium uptake, indicating protection induced by the β T1A substitution compared to the 20S_{WT} variant. Notably, this protective effect in response to the β T1A substitution extends across a wider range of β -subunit peptides at the α - β interface (residues β 85-91, β 87-91, β 87-98, β 99-104, β 99-110) compared to the fewer peptides affected as a result of ixazomib binding (residues β 87-98).”

- P9.L241: *“and the structural elements involved in RP binding” please highlight these residues in Extended Data Fig.6a.*

This is an excellent suggestion. We have annotated the figure as per the reviewer’s recommendation.

- P10.L246-257: *The results observed on the OG mutant are striking and make sense. However, the increased deuteration is explained here only as a propagation of conformational changes, whereas the physical removal of the gates probably eases the diffusion of deuterons inside the 20S cavities. Therefore, an increased accessibility to the (deuterated) solvent could also be partly responsible for this increased deuteration.*

We appreciate the reviewer’s point regarding the observed D-uptake in the OG variant. This point to some extent is related to the one regarding the diffusion of substrate that we addressed above. However, we believe that it is unlikely that increased deuteration might be primarily due to easier diffusion of D₂O inside the 20S cavity due to the physical removal of gating residues. The proteasome chamber is not sealed, and the short dimensions of the degradation chamber ensure that solvent accessibility is not a limiting factor for deuteration. In fact, our tripeptide substrate, significantly bigger than water, diffuse into the 20S CP without impediment, as demonstrated in our substrate-binding studies. Furthermore, the persistent differences in deuteration patterns we observe for several hours in our experiments argue against a simple diffusion-based explanation. These sustained differences are indicative of genuine conformational changes propagated by the removal of the gating residues, rather than merely enhanced solvent access. We have added the following to the text to clarify our interpretation of our HDX-MS data:

“The reaction progress curves for both the 20S_{WT} and 20S_{OG} variants display a rapid initial rate and are sensitive to changes in initial substrate concentration

(Supplementary Fig. 4a, b)⁴³. These observations and the microscopic length scales involved (Supplementary Fig. 4c) suggest that the changes in kinetic parameters arise from allosteric modulation of the 20S proteasome due to gate removal, rather than from decreased steric hindrance caused by the absence of gating residues. Nonetheless, we concede that the role of gating residues as a diffusion barrier may also influence these changes.”

and

“While the physical removal of the gating residues might facilitate easier access for D₂O, the significant and sustained differences in deuterium uptake patterns and efficient diffusion over microscopic length scales are indicative of conformational changes in 20S CP rather than merely increased solvent accessibility.”

• *P11.L259: While the Figures of the manuscript are generally very well designed, the Reviewer is wondering about the impact/efficiency of Fig.3a. The top part of the panel seems unnecessary (since the bottom part is enough and self-explanatory). However only the timepoint at 10sec is represented here. The Reviewer suggests to remove the top part of panel a and replace it by the heatmaps of Extended Data Fig6.*

We fully agree with this reviewer that our Fig 3a was not effective in communicating our HDX-MS data. As per the reviewer’s suggestion, we have moved Extended Fig 6 and 7 to the main text. They now appear as Figures 3 and 4.

• *P13.L294: “This motion is present in each 20S variant, except for the ixazomib-bound 20SWT variant, suggesting that inhibitor binding may limit the overall mobility of the 20S CP.” Could this be related to the differences in local resolution (Extended Fig.5) that show indeed some “rigidification” of the ixazomib bound 20S, compared to the WT and OG mutants? However, the resolution is even better for the T1A mutant, but the motion described in the 3DVA analysis is nonetheless present...Could the author comment on that?*

We appreciate the reviewer’s comment on the correlation between the observed motion and the differences in local resolution across the 20S variants. The absence of the specific motion in the ixazomib-bound 20SWT variant can indeed be related to differences in local resolution and the apparent “rigidification” of this variant compared to the WT and OG variant (Supplementary Movies 1 – 6). However, it is important to clarify that a direct correlation between the motions identified by 3DVA and local resolution cannot be established due to fundamental differences in what these approaches measure. 3DVA is designed to deconvolute the structural heterogeneity present in cryo-EM datasets into distinct modes of motion, referred to as principal components. Each principal component represents a specific type or direction of motion, such as rotations, translations, or expansions, which are independent of each other. While 3DVA provides a qualitative representation of these motions, it does not quantify their absolute contributions to the overall structural heterogeneity. In contrast, local resolution represents an averaged

measure of structural rigidity and disorder, integrating the effects of all motions simultaneously. This distinction is critical when interpreting the observed rigidification in the ixazomib-bound 20SWT variant. From 3DVA, we can identify and describe a specific principal component representing the motion reduced by ixazomib binding. However, other motions not captured by this principal component may still persist and contribute to the overall heterogeneity of the dataset. These remaining motions could limit the extent of local resolution improvement in the ixazomib-bound 20SWT, even though the specific motion described in our analysis is suppressed. This may explain why the ixazomib-bound variant has a lower resolution compared to the T1A mutant, despite appearing more rigidified in certain respects. Overall, compared to the WT and OG mutants, the ixazomib-bound 20SWT appears more rigid, as observed in the local resolution maps. However, we acknowledge that this observation and the associated terminology could be confusing. To clarify this in the manuscript, we have revised the relevant text in the Results section as follows:

“Rapid freezing of protein samples for cryo-EM preserves structural variations which can be uncovered using 3DVA. 3DVA deconvolutes structural heterogeneity in cryo-EM datasets into distinct modes of motion, known as principal component, each ideally representing a unique and independent conformational change, such as rotation, translation, or expansion. By identifying a principal component of interest, particles can be grouped along the principal component axis into subsets, which are then refined to produce high-resolution maps of the different conformations describing subtle and distinct structural motions⁵². 3DVA of our 20S CP datasets revealed three distinct motions within our maps.”

and

“A similar motion is present in each 20S variant, except for the ixazomib-bound 20S_{WT} variant, suggesting that inhibitor binding may reduce this type of motion in the 20S CP.”

• *P14.L335-371: The analysis of the different mutants corroborates the proposed allosteric mechanism. One might then wonder how would the dynamic “breathing” of these constructs be affected. Would it be exacerbated in the alphaK52F/betaV53Q double mutant and strongly reduced in the betaY35F mutant? Although such experiments probably go beyond the scope of this article, could the authors propose working hypotheses?*

We appreciate the reviewer’s interest in the dynamic “breathing” of these constructs and how it might vary among the variants. While intriguing, we are hesitant to comment on this as it indeed beyond the scope of manuscript and would be highly speculative without direct experimental evidence.

• *P14.L335-371: Did you try to combine these mutations with the original OG and betaT1A variants and measure their activity?*

The T1A variant inherently lacks catalytic activity as its nucleophilic Thr has been replaced with an Ala residue. Therefore, it is unsuitable for the suggested experiments. However, we did test that these two alpha and beta subunit variants assemble into 20S CP and added this to the Methods section:

“The 20S CP assembly yield was the highest for 20S_{WT}, followed by 20S_{βT1A}, 20S_{αΔ2-6}. The αΔ2-6, βT1A construct also assembled into mature 20S particles, albeit with substantially lower yield.”

• P17.412-416: *“Molecules that selectively bind to the inactive conformation (20SOFF) of the proteasome established in this study, rather than the well-known active form (20SON) could be critical to future inhibitor development efforts that focus on allosteric rather than orthosteric sites of the proteasome in bacterial and eukaryotic systems.” Do you suggest as an SAR strategy to stabilize the inactive 20S form by filling up the S1 pocket or by inducing this conformational change of the Switch Helix I? Given the presence of three different catalytic subunits in eukaryotic 20S, do you think that it would be possible to target all three subunits concomitantly with the same drug, or that impairing just one would be enough?*

To clarify, our suggestion was aimed at the development of compounds that stabilize the 20S_{auto-inhibited} conformation of the proteasome, rather than compounds that specifically occupy the S1 pocket. The text has been modified to reflect this:

“Our structural insights advance our understanding of the functional dynamics of *Mtb* 20S CP. Molecules that selectively bind to and stabilize the 20S_{auto-inhibited} of the proteasome established in this study, rather than the well-known canonical resting state (20S_{resting}) could be critical to future inhibitor development efforts that focus on allosteric rather than orthosteric sites of *Mtb* 20S CP.”

Regarding the second part of your question, given the differences between bacterial and eukaryotic proteasome structures that reviewer #1 also referred to, particularly in the composition of catalytic subunits in eukaryotic 20S proteasomes, it would be speculative to discuss the feasibility of targeting all three subunits concomitantly or the efficacy of impairing just one subunit. We have accordingly revised our manuscript to emphasize that our current findings pertain specifically to the *Mtb* 20S proteasome and avoid broader speculation on the eukaryotic system, except when supported by specific references:

“Aspects of the mechanism described here appear to be conserved in other structurally homologous systems. For example, allosteric activation of HsIV protease, the *E. coli* homologue to the proteasome and the ancestor of archaeobacterial and eukaryotic 20S proteasomes⁵⁷, involves the rotation of switch helices to correctly position the active site (Fig. 7), similar to what Losick and colleagues have proposed for the protein phosphatase 2C superfamily⁴². Interestingly, the binding of RPs or substitutions at the α-β interface of the archaeal *T. acidophilum* 20S CP modulate substrate specificity and alter the

cleavage products³⁷. These likely result from changes in the topology of the substrate binding pocket of the 20S CP due to the movement of the switch helices in that organism. Therefore, the mechanism described here may also apply to other structurally homologous systems, though, this needs to be experimentally tested.”

Figure 7. Switch helices are conserved and regulate activity in the ancestral enzyme, HslV. Comparison of the switch helices from the 20S β -subunit to those from HslV, an ATP-dependent protease homologous to the proteasome β -subunit, highlight conserved structural transitions that regulate proteolytic function. The C-terminal region of switch helix II undergoes an order to disordered conformational change between inactive (PDB 1G3K) and active (PDB 1G3I) states of HslV in a manner similar to that seen in the *Mtb* 20S β -subunit.

- P18.L456: The *MassIVE* repository access number should be stated for proper reviewing.

We apologize for this oversight. The *MassIVE* access number is now included in the manuscript.

- P26.L674: “Initial velocities were extracted from the first 30 seconds of each reaction using Python scripts written in-house”. Did you check, or does your Python scripts take into account the linearity of the slope to measure the activity? Some lag in this linear region might appear, which may affect the calculated values.

Our peptidase assays exhibited linear initial velocities without any lag phase. Our in-house Python script specifically assesses and confirms the linearity of these reaction progress curves before extracting initial slopes. We have added representative progress curves in Supplementary Figure 4.

Supplementary Figure 4. Representative progression curves reveal that the gating residues do not inhibit Z-VLR-AMC degradation. (a) Reaction progression initially increases at a rapid rate and then deaccelerates for both 20S_{WT} and 20S_{OG} variants; (b) A 30-sec fitting window was used to determine the initial velocities of substrate degradation. The calculated fits used to determine the rate of product formation are shown as black lines; and (c) The root mean square distance $(\langle x^2 \rangle)^{1/2}$ traveled by diffusing particles in time t , estimated using the Einstein-Smoluchowski equation. We used a diffusion coefficient D of $3.2 \times 10^{-10} \text{ m}^2\text{s}^{-1}$, estimated for Z-VLR-AMC using HydroPro²⁵ at room temperature, being reduced by 10-, 100-, and 1000-fold by the gating residues; and (d) The concentration profiles above a plane where a solute is diffusing. The curves represent the equation:

$$c(x, t) = \frac{n_0}{A(\pi Dt)^{1/2}} e^{-x^2/4Dt}$$

Where each curve is labeled with different values of Dt . The units of Dt (displayed on the plot for each curve) and x are arbitrary but chosen such that Dt/x^2 remains dimensionless. For instance, if x is measured in meters, Dt would

be in square meters. On the microscopic length scale, for an ordinary diffusion coefficient of $D = 10^9 \text{ nm}^2 \text{ s}^{-1}$, a Dt value 1 nm^2 corresponds to $t = 1 \text{ s}$. These plots demonstrate the effectiveness of diffusion over microscopic length scales. See ref²⁶.

- *P30.L782: The authors did inject some GdHCl to clean the sample loop but it seems that they did not run a proper blank (same chromatographic run) between each injection, which is a good practice to avoid carry-over on the C18 column. This is particularly crucial when EX1 or mixed EX1-EX2 are observed, to avoid mis-interpretation of the dataset. The fact that most peptides do not show this mixed behavior is not enough, since not all peptides bind the column with the same affinity. Did the authors make sure beforehand that no carry-over was present in these conditions?*

The reviewer correctly raises a point regarding the importance of minimizing column-carry-over during HDX-MS experiments. We implemented a stringent protocol to ensure our HDX-MS data are free from column carry-over. Each injection is followed by several flushes of the injection port with a solution containing 3 M GdnHCl and 1.5 mM n-dodecylphosphocholine, which effectively removes residual hydrophobic and sticky peptides left in the sample port. Additionally, the protease column, which we identified as the major source of column carry-over, was flushed several times with a full loop of the cleaning solution after the conclusion of peptide trapping and as the analytical run was in progress. This cleaning routine proved effective in reducing column carry-over to undetectable levels for our proteasome samples, as judged by blank injection using the same inlet and MS methods. We have detailed these procedures in the Methods section and have now emphasized key steps to clarify our approach in response to your comment:

“Liquid handling and reverse-phase separation was performed using a M-Class nanoAcquity UPLC system equipped with HDX technology (Waters, Milford, MA, USA). 5 pmol of sample was digested at 15 °C using a nepenthesin-2 column (Affipro, AP-PC-004, 1 mm × 20 mm, 16.2 μL). The resulting peptides were trapped for 3 min on a BEH C18 (1.7 μm, 2.1 mm × 5 mm; Part#: 186003975, Waters) column at a flowrate of 100 μL min⁻¹. Peptide separation was achieved on an HSS T3 (1.8 μm, 1.0 × 50 mm; Part#: 186003535, Waters) column using a linear, 8-minute acetonitrile:H₂O gradient acidified with 0.1% formic acid (acetonitrile ramped from 5 – 35%) at 0 °C and at a flowrate of 100 μL min⁻¹. Between each injection, the sample loop and the protease column were flushed using 1.5 M guanidine hydrochloride, 4% (v/v) acetonitrile, 0.8% (v/v) formic acid, and 1.5 mM n-dodecylphosphocholine. This routine proved effective in reducing column carry-over to undetectable levels for our proteasome samples, as judged by blank injection using the same inlet and MS methods.”

Minor comments:

P3.L57: In legend of Fig1; the authors refer to the PDB 3M10, whereas the latter corresponds to a substrate-free form of Arginine Kinase...In panel c, the authors refer to structures on the left (CP-WT) and on the right (CP-OG), however the structure on the

right (cartoon) is part of panel d...Is this an error or are the authors talking about the structure from panel d ? Please clarify

We apologize for this oversight and have corrected this in the manuscript. This should have read 3MI0 and not 3M10.

P4. L73: please remove “recently”, since the ref is almost 10 years old.

The word “recently” has been removed as per the reviewer’s recommendation.

Extended Data Fig. 4 is cited in the text before extended Data Fig. 3.

We have ensured that the figures are called out to in the text in order.

P13.L319: “Concurrently, residues 92-98 of switch helix II show increasing order, with their density becoming well-defined by frame 20” suggestion: “explaining the slower deuteration uptake of peptide 87-98 (Extended Data Fig.6).

We have incorporated this suggestion into our manuscript.

P.28L744: “These sorted symmetry expanded particles were...”

We have corrected this grammatical error and thank the reviewer for pointing this out.

Extended Data Figures:

P3.L846: please check ref. 3.

The reviewer is absolutely correct, and we appreciate the attention to detail. We mistakenly cited the wrong paper in reference 3. We have now corrected it to:

G. Hu et al., "Structure of the *Mycobacterium tuberculosis* proteasome and mechanism of inhibition by a peptidyl boronate," *Mol. Microbiol.* 59, 1417–1428 (2006).

Thank you for pointing this out.